# DeepImageSearch: Benchmarking Multimodal Agents for Context-Aware Image Retrieval in Visual Histories

**Chenlong Deng**[1][*] **Mengjie Deng**[1][*] **Junjie Wu**[2] **Dun Zeng**[2] **Teng Wang**[2] **Qingsong Xie**[2] **Jiadeng Huang**[2]
**Shengjie Ma**[1] **Changwang Zhang**[2] **Zhaoxiang Wang**[2] **Jun Wang**[2] **Yutao Zhu**[1] **Zhicheng Dou**[1]

## Abstract

Existing multimodal retrieval systems excel at semantic matching but implicitly assume that query-image relevance can be measured in isolation. This paradigm overlooks the rich dependencies inherent in realistic visual streams, where information is distributed across temporal sequences rather than confined to single snapshots. To bridge this gap, we introduce DeepImageSearch, a novel agentic paradigm that reformulates image retrieval as an autonomous exploration task. Models must plan and perform multi-step reasoning over raw visual histories to locate targets based on implicit contextual cues. We construct DISBench, a challenging benchmark built on interconnected visual data. To address the scalability challenge of creating context-dependent queries, we propose a human-model collaborative pipeline that employs vision-language models to mine latent spatiotemporal associations, effectively offloading intensive context discovery before human verification. Furthermore, we build a robust baseline using a modular agent framework equipped with fine-grained tools and a dual-memory system for long-horizon navigation. Extensive experiments demonstrate that DISBench poses significant challenges to state-of-the-art models, highlighting the necessity of incorporating agentic reasoning into next-generation retrieval systems.

## 1. Introduction

Image retrieval serves as a fundamental mechanism for information access, allowing users to efficiently locate visual

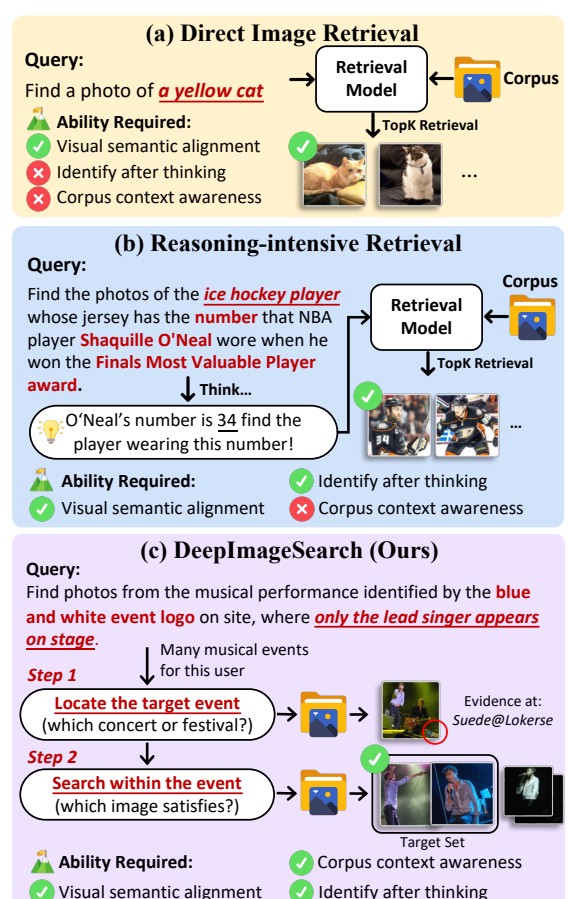

*Figure 1.* **Evolution of image retrieval paradigms.** (a) Direct retrieval matches queries to images through visual semantic alignment. (b) Reasoning-intensive retrieval requires inference over external knowledge, but still evaluates each image independently. (c) DeepImageSearch demands corpus context awareness, where models must first locate target events within the visual history and then identify qualifying images through multi-step reasoning.

content within large repositories ranging from web-scale databases to personal photo albums (Radford et al., 2021; Jia et al., 2021). With the development of vision-language models, this field has witnessed great progress, where models demonstrate stronger and stronger capabilities in aligning items in various modals (Jiang et al., 2025; Zhou et al., 2025b). The dominant paradigm underlying these models relies on *independent instance matching*. It measures the

---

[*]Equal contribution   [1]Gaoling School of Artificial Intelligence, Renmin University of China, Beijing, China [2]OPPO Research Institute. Correspondence to: Chenlong Deng <dengchenlong@ruc.edu.cn>, Zhicheng Dou <dou@ruc.edu.cn>.

*Proceedings of the 43rd International Conference on Machine Learning*, Seoul, South Korea. PMLR 306, 2026. Copyright 2026 by the author(s).

[-1]Our code, data and leaderboard are available at https://github.com/RUC-NLPIR/DeepImageSearch.

semantic relevance between the query and each candidate image in isolation, without considering other images. By computing similarity scores for every image individually, models retrieve the candidates that best correspond to the user input. This simple yet effective approach establishes the backbone for modern visual search applications.

However, this paradigm faces significant challenges when handling complex user intents that cannot be fully captured by a single visual embedding, particularly in personal albums where users rely on contextual cues such as events, time, and companions to locate specific memories (Naaman et al., 2004; Whittaker et al., 2010). Recent reasoning-intensive (Cui et al., 2025) or agentic approaches (Cheng et al., 2025a; Tu et al., 2025) address this limitation by employing language models to decompose queries or incorporate external knowledge to refine the search target before matching (as shown in Figure 1 (b), where the model infers a specific jersey number from a player's name). While these methods improve semantic understanding, they still operate under the independent matching assumption, *i.e.*, the target image can be identified in isolation once the textual intent is clarified. In reality, user intents often require information inherently distributed across temporal or causal image sequences, requiring reasoning that interleaves textual and visual evidence. For example, as illustrated in Figure 1 (c), a user is searching for concert photos where *"only the lead singer appears on stage"*. The user's collection may contain numerous visually similar concert photos, and the target images themselves lack distinctive features to verify the specific event. However, if the user recalls a *"blue-and-white event logo"* at the venue, this clue serves as an anchor to first identify the correct event, after which the targets can be located. Crucially, the evidence required to resolve this query (the logo) and the target (the singer) appear in different images. Since the target cannot be distinguished by appearance alone, models must perform **corpus-level contextual reasoning**: actively exploring and chaining scattered visual evidence within the corpus. Unlike static knowledge retrieval, this dynamic capability remains largely unexplored by existing benchmarks.

To address this limitation, we propose **DeepImageSearch**, a novel paradigm that reformulates image retrieval as an agentic exploration task. Different from conventional methods that use a static corpus for passive ranking, DeepImageSearch requires models to discover latent logical structures within the data itself through active exploration. Under this paradigm, models must autonomously plan search trajectories, coordinate fine-grained perception tools, and connect scattered clues to build evidence chains. This transforms retrieval from a one-shot matching task into a multi-step reasoning process, bridging the gap between complex user queries and precise targets.

Constructing such benchmarks presents a significant scalability challenge: creating context-dependent queries requires annotators to identify subtle cross-event connections within massive collections, causing a significant cognitive load for manual processing. To address this, we introduce a human-model collaborative pipeline that offloads the labor-intensive context discovery to models. Specifically, we employ vision-language models to parse visual histories and construct a spatiotemporal memory graph, automatically mining recurrent clues and potential reasoning paths. These candidates are then verified and refined by human annotators. This approach yields DeepImageSearch-Bench (abbreviated as DISBench), the first large-scale benchmark dedicated to this task, effectively balancing reasoning depth with annotation efficiency.

To support future research on this task, we design a baseline agent framework with tools and memory mechanisms tailored for visual history exploration. Benchmarking state-of-the-art multimodal models on DISBench reveals a significant performance gap. The best-performing model achieves an EM-score of only 28.7, in contrast to near-ceiling results on conventional retrieval benchmarks. Error analysis indicates that current models still struggle with long-horizon exploration, frequently losing track of reasoning states or failing to discover cross-event associations. These findings confirm that corpus-level contextual reasoning remains a critical unsolved problem, positioning DISBench as a valuable testbed for advancing this direction.

Our contributions can be summarized as follows:

1) We propose DeepImageSearch, a novel paradigm that reformulates image retrieval from independent matching to context-dependent reasoning over visual histories.
2) We construct DISBench, the first benchmark for this task, constructed via a human-model collaborative pipeline that ensures both reasoning depth and data quality.
3) We develop a specialized agent framework and conduct extensive experiments, revealing critical capability gaps in long-horizon exploration and establishing a robust baseline for future research.

## 2. Related Work

### 2.1. Multimodal Retrieval and Benchmarks

The evolution of multimodal representation learning, from basic vision-text alignment to advanced multimodal foundation architectures, has reshaped retrieval technology (Radford et al., 2021; Jia et al., 2021; Li et al., 2022; Zhai et al., 2023; Zhang et al., 2024a; Wei et al., 2024; Lin et al., 2025; Zhang et al., 2024b; Zhou et al., 2025b; Chen et al., 2025b;a; Zhu et al., 2023). To evaluate these capabilities, extensive benchmarks have been established with evaluation scope expanding from pure semantic matching to diverse scenarios

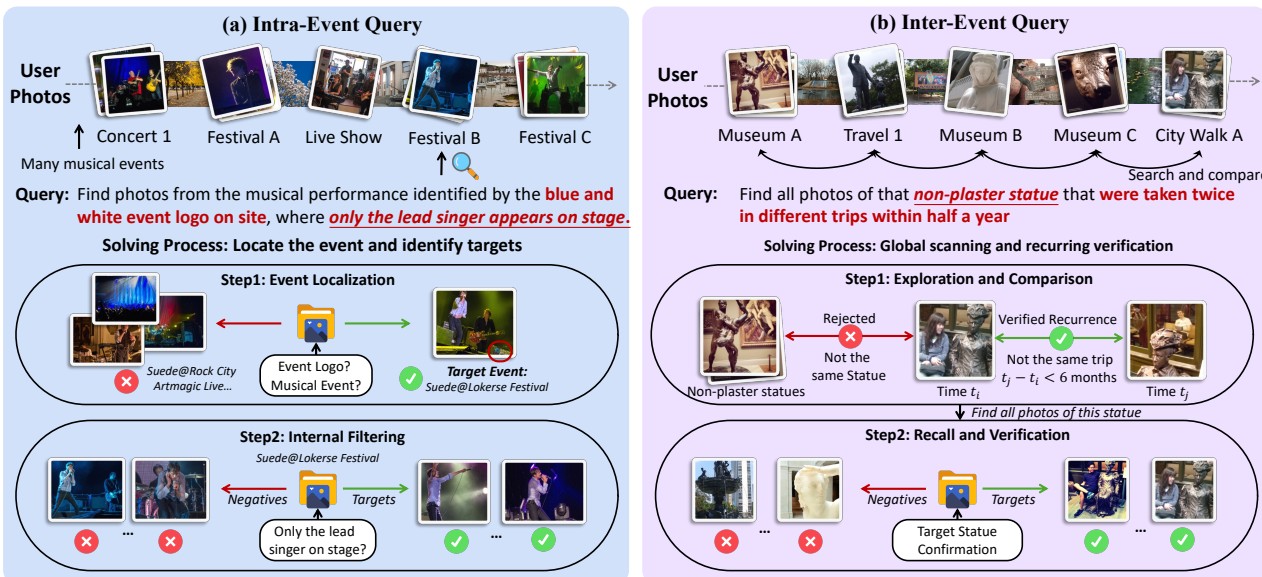

*Figure 2.* Two query types in DISBench. (a) Intra-Event queries locate a specific event and filter targets within it. (b) Inter-Event queries scan across events to verify recurring elements under temporal/spatial constraints.

such as complex compositional understanding and temporal video retrieval (Jiang et al., 2025; Faysse et al., 2025; Meng et al., 2025; Zhou et al., 2025a; Ning et al., 2026). However, most existing benchmarks evaluate query-target relevance independently, overlooking structured associations within the data. DeepImageSearch addresses this limitation by requiring models to perform corpus-level contextual reasoning over raw visual histories.

## 2.2. Benchmarking Multimodal Agents

Multimodal agents have demonstrated strong planning and reasoning capabilities across complex interactive tasks (Yin et al., 2023; Yao et al., 2025). Correspondingly, a series of benchmarks have been proposed to evaluate these capabilities, spanning web search (Li et al., 2025), GUI manipulation (Zhou et al., 2024; Deng et al., 2023; Gou et al., 2025; Xie et al., 2024), gaming (Zheng et al., 2025), and embodied intelligence (Cheng et al., 2025b), which have effectively driven rapid progress in this field. However, they have not explored image retrieval settings that inherently demand agentic reasoning. DeepImageSearch represents a new stage of image retrieval where targets cannot be identified without multi-step exploration over corpus context, making agentic capabilities essential rather than auxiliary.

## 3. DISBench: The Proposed Dataset

### 3.1. Task Formulation

We formalize DeepImageSearch as a context-aware set retrieval task. Given a user's visual history $\mathcal{C} = \{I_1, I_2, ..., I_N\}$ ordered chronologically, each image $I_i =$

$(v_i, m_i)$ contains visual content $v_i$ and metadata $m_i$ including timestamps and GPS coordinates. Upon receiving a natural language query $Q$, the system predicts a target subset $\mathcal{R} \subseteq \mathcal{C}$ containing all images that satisfy $Q$. Unlike conventional retrieval that independently scores each image, our task requires modeling $P(\mathcal{R}|Q, \mathcal{C})$, where the relevance of each image may depend on other images in $\mathcal{C}$.

All queries in our benchmark are text-only, but this design covers scenarios where users provide reference images. We convert such visual references into textual descriptions, requiring models to first locate these visual anchors within the corpus before performing spatiotemporal reasoning. For instance, a query mentioning "the concert with a blue-and-white logo" requires the model to first retrieve images containing this logo, then use them as anchors for reasoning. This increases task difficulty by preventing models from bypassing the exploration through direct visual matching.

### 3.2. Dataset Criteria and Source

Our task evaluates corpus-level contextual reasoning, which requires models to discover latent associations across images to resolve queries that independent semantic matching cannot handle. Based on this core capability, we categorize queries into two types as illustrated in Figure 2. Intra-Event queries require first locating a specific event and then filtering target images within it. For example, a user searching for concert photos with only the lead singer on stage must first identify the correct concert via a remembered logo, then select qualifying images from that event. Inter-Event queries demand scanning across multiple events to find recurring elements that satisfy temporal or spatial constraints. For

instance, finding all photos of a statue that appears in different trips within half a year requires comparing candidates across the timeline and verifying their recurrence.

Supporting these two query types imposes requirements on data characteristics. The corpus must exhibit temporal continuity and user-centric coherence where the same entities recur across events, enabling both event-level localization and cross-event association discovery. Unlike retrieval benchmarks that aggregate discrete images from diverse sources, we construct our benchmark from YFCC100M (Thomee et al., 2016), which naturally preserves a hierarchical structure of users, photosets, and photos. A photoset refers to a collection of photos grouped by users during upload, typically corresponding to a single event such as a concert or trip. This structure provides ground-truth event boundaries for automated query construction but remains completely invisible to models during evaluation, forcing them to discover events autonomously. We accumulate complete photosets until reaching a capacity of 2,000 photos per user, simulating realistic memory search over several years of visual history while ensuring sufficient scale for cross-event associations.

### 3.3. Context Mining and Query Synthesis

Constructing context-dependent queries requires identifying connections across thousands of images, which imposes substantial cognitive load on human annotators. To address this challenge, we propose a semi-automated pipeline that offloads context discovery to vision-language models while reserving human effort for verification. As illustrated in Figure 3, our pipeline consists of four stages. We describe each stage below and provide details in Appendix C.

**Visual Semantic Parsing.** We first employ a vision-language model to parse each image with its metadata and photoset context. The model extracts visual cues, defined as specific entities that characterize a scene's uniqueness, such as distinctive landmarks, salient objects, or visible text. For person identity, we use face detection and clustering to track recurring individuals, then prompt the model to describe each person's attributes in the current image. This stage produces structured descriptions for each image, including visual summaries, visual cue lists, and person states, which serve as raw materials for mining cross-image associations.

**Latent Association Mining.** The core challenge is discovering latent associations among extracted clues across space and time. We propose an efficient retrieval-verification pipeline to avoid exhaustive pairwise comparison. For each visual clue, we encode its source image and textual description into a query vector, then retrieve top-k candidates from both within and outside the source photoset. This hybrid strategy ensures that long-range associations are not overshadowed by short-term visual recurrence. Retrieved

candidates proceed to verification, where a vision-language model determines whether each candidate image contains the same visual clue as the source. This step filters out false positives that are visually similar but semantically unrelated, yielding high-confidence association links.

**Memory Graph Construction.** We organize the extracted entities and verified associations into a heterogeneous memory graph $\mathcal{G} = (\mathcal{V}, \mathcal{E})$. (1) The graph contains four node types: Photo nodes as atomic visual units, Photoset nodes for event-level context, Visual Clue nodes for fine-grained entities, and Person nodes for human identities tracked via face clustering. (2) The edge set $\mathcal{E}$ contains two categories of relationships. Structural edges connect each Photoset to its contained Photo nodes and link each Photo to its associated Visual Clue and Person nodes, capturing membership and containment relations. Association edges connect Visual Clue nodes directly to target Photo nodes where the same entity reappears, with each edge carrying a natural language description that explains the connection rationale. This structure exhibits a connectivity pattern: nodes within the same event are naturally clustered through their shared Photoset node, while cross-event connectivity relies entirely on association edges between Visual Clues. The memory graph thus explicitly captures the fragmented nature of visual history and establishes the topological foundation for sampling cross-event reasoning paths in subsequent stages.

**Subgraph Sampling and Query Synthesis.** The full memory graph is too large for direct query construction, so we sample meaningful local subgraphs. Starting from a randomly selected photo node, we iteratively expand the subgraph by uniformly sampling edge types and then edges of the chosen type. This strategy balances intra-event density with cross-event associations. Sampling terminates when edge count reaches a predefined limit, after which we add any missing photoset nodes to ensure event context completeness. For query synthesis, we serialize sampled subgraphs into structured text containing node attributes and association rationales. We prompt vision-language models to construct queries requiring multi-step reasoning over event context, person identity, and object features. Query construction enforces that targets exhibit visual ambiguity, with identifiability stemming from contextual associations rather than distinctive appearance. We also retrieve external knowledge for the mentioned entities to generate paraphrased alternatives for human verification.

### 3.4. Human Verification and Refinement

We assemble an annotation team of seven computer science professionals, all holding master's degrees or above. Annotators work with a dedicated retrieval interface that supports multimodal search, temporal and spatial filtering, and event-

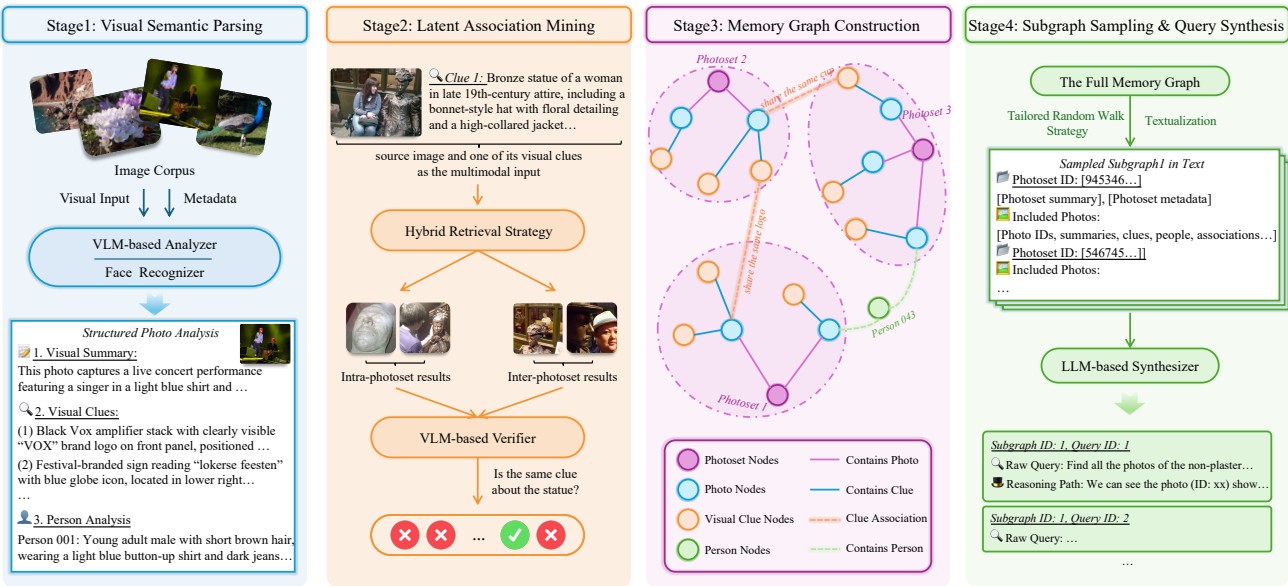

*Figure 3.* **Semi-automated data construction pipeline.** Starting from raw images, we first parse visual content to extract salient clues and person attributes, then mine latent associations across the corpus through a retrieval and verification strategy. These elements are organized into a memory graph, from which we sample subgraphs via random walks to synthesize candidate queries for human verification.

based browsing. The verification process consists of four stages. **(1) Quality Filtering.** Annotators first examine the correctness of candidate queries by verifying that the referenced clues exist and the reasoning chains are logically valid. Annotators then assess difficulty to ensure that queries cannot be solved through direct semantic matching. Specifically, we require that the corpus contains visually similar distractors that are indistinguishable from targets based on appearance alone, and that resolving this ambiguity requires contextual reasoning over events or temporal information. This criterion enforces multi-step exploration rather than single-shot retrieval. Applying these strict criteria, we retain 122 queries from 2,000 candidates, yielding a retention rate of 6.1%. **(2) Exhaustive Target Annotation.** For retained queries, annotators identify all qualifying images through systematic exploration. They employ multimodal retrieval to locate visually similar candidates across the corpus, then apply metadata filters and event-level examination to verify whether each candidate satisfies the query constraints. This process ensures that no qualifying images are overlooked due to their visual similarity to non-targets. **(3) Language Refinement.** Annotators further refine the query language to improve naturalness and fluency. When necessary, they paraphrase entity references or restructure descriptions to increase retrieval difficulty while preserving the original intent. **(4) Cross-Validation.** To ensure annotation quality, different annotators independently label the target image set for the same query. We compute the Intersection over Union (IoU) between their annotations to measure agreement. The average IoU reaches 0.91, indicating high consistency. All divergent cases are resolved through joint discussion among the two original annotators and a third annotator.

### 3.5. Dataset Statistics

Figure 4 summarizes the statistics of DISBench. The benchmark contains 122 queries distributed across 57 users and 109,467 photos, with each user's visual history spanning 3.4 years on average. Each query targets 3.84 images on average, and models must identify all qualifying images without prior knowledge of the expected count. We categorize queries into two types based on their reasoning patterns. Intra-Event queries (46.7%) require first locating a specific event through contextual clues and then retrieving target images within it. Inter-Event queries (53.3%) demand collecting and comparing evidence across multiple distinct events. Both types fundamentally differ from traditional retrieval paradigms, as neither can be solved through direct semantic matching between queries and individual images. Figure 4(b) shows that target images span diverse themes including portraits and people (41.8%), nature views (18.9%), daily items (14.8%), and scenic spots and architectures (11.5%), reflecting the variety of real-world visual memories. Due to the open-ended nature of agentic reasoning, multiple valid paths may exist for a single query, and we therefore do not prescribe fixed reasoning steps.

## 4. ImageSeeker: An Agentic Framework

To support research on this new task, we design Image-Seeker, a simple yet effective baseline agent framework. This task presents unique challenges that inform our design. First, agents must explore large photo collections through a combination of semantic retrieval, metadata reasoning, and visual verification, requiring a coordinated

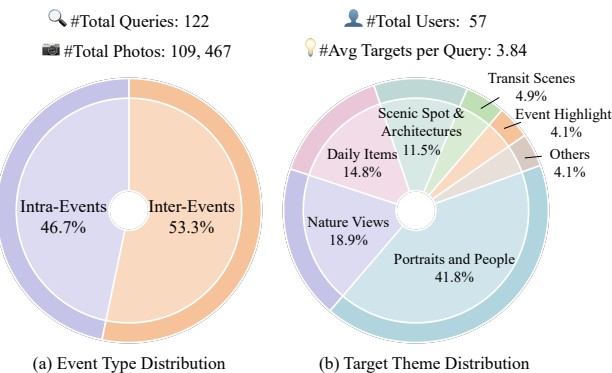

(a) Event Type Distribution  (b) Target Theme Distribution

*Figure 4.* **Dataset statistics of DISBench.** (a) Query type distribution shows a balanced split between intra-event and inter-event queries. (b) Target images span diverse themes, including portraits, nature views, daily items, and scenic spots.

tool set. Second, answering a single query may involve many interaction steps, and processing numerous images can quickly exhaust context limits, necessitating memory mechanisms that maintain reasoning state while managing context length. Our framework addresses these challenges through targeted tool and memory design, providing insights for future work. Planning is handled through structured prompting that guides query decomposition and constraint identification. We describe each component below and provide full implementation details in Appendix D.

**Tools for Visual History Navigation.** Exploring visual histories requires three core capabilities: retrieving relevant photos from large collections, leveraging metadata for precise constraints, and performing fine-grained visual verification. We design tools for each capability. For retrieval, ImageSearch accepts text, images, or interleaved queries and returns semantically similar photos. For metadata operations, GetMetadata reads timestamps and addresses of specified photos, and FilterMetadata selects photos satisfying temporal or spatial conditions. For visual verification, ViewPhotos injects photos into the agent's context for direct inspection. We additionally provide WebSearch for resolving external entities that may appear in user queries.[0] Since no single tool can answer complex queries, agents must combine these operations to construct reasoning paths. To support this, ImageSearch and FilterMetadata allow agents to save results as named photo subsets and to perform subsequent retrieval or filtering within these subsets. For example, an agent can first filter all photos from a specific month, save them as a subset, and then search within this subset for photos containing a particular object.

**Memory Mechanisms.** We design two complementary memory mechanisms to support multi-step reasoning. The first is **explicit state memory** through photo subsets. Since

queries require iterative exploration where each step builds on previous discoveries, agents need to persist intermediate results. As described above, agents can save retrieval or filtering results as named variables. These subsets persist across reasoning steps and support operations such as constrained search and intersection, allowing agents to narrow down candidates. The second is **compressed context memory**. As reasoning paths in this task can span many steps, the growing interaction history and large image quantities may exceed context limits. We address this by summarizing the history into a three-tier structure when triggered by a token length limit, an image count limit, or proactive agent invocation to condense trial-and-error history upon detecting a mistake. Specifically, *global* memory preserves high-level goals and key findings, *local* memory records the current subgoal and plans, and *tool-use* memory summarizes failure experiences to prevent repeated mistakes. This separation maintains global direction, local state, and operational efficiency under strict context constraints.

## 5. Experiments

### 5.1. Experimental Setup

**Agentic Evaluation.** We evaluate state-of-the-art multimodal models as agents using our proposed ImageSeeker framework. Our evaluation covers leading proprietary models including GPT-4o (Hurst et al., 2024), GPT-5.2 (OpenAI, 2025), Gemini-3-Flash-Preview, Gemini-3-Pro-Preview (Google, 2025), Claude-Sonnet-4.5 (Anthropic, 2025b), and Claude-Opus-4.5 (Anthropic, 2025a), as well as open-source models including Qwen3-VL-235B-A22B-Thinking, Qwen3-VL-235B-A22B-Instruct, Qwen3-VL-32B (Bai et al., 2025) and GLM-4.6V (Team et al., 2025). All models are equipped with identical tool interfaces and memory mechanisms, isolating the backbone's planning and reasoning ability as the primary variable. We adopt Exact Match (EM) and F1 as evaluation metrics. Detailed configurations and hyperparameters are provided in Appendix D.4.

**Retrieval Evaluation.** To demonstrate the limitations of conventional retrieval paradigms, we also evaluate three representative vision-language embedding models: Qwen3-VL-Embedding (2B and 8B) (Li et al., 2026) and Seed-1.6-Embedding (ByteDance Seed, 2025). For each query, we encode the text and retrieve images from the corpus based on cosine similarity. We report MAP@$k$, Recall@$k$ and NDCG@$k$ with $k \in \{1, 3, 5, 10\}$.

**In-depth Analysis.** For ablation studies, test-time scaling, and error analysis, we use Gemini-3-Flash-Preview with Qwen3-VL-Embedding-8B as the representative configuration, which balances performance and cost efficiency.

---

[0] Text in blue box refers to the tool name.

*Table 1.* Comparison of different models with the ImageSeeker framework on DISBench. **Bold** and underline indicate the best and second-best results, respectively.

| Base Model | Qwen3-VL-Embedding-2B | | | | | | Qwen3-VL-Embedding-8B | | | | | |
| --- | --- | --- | --- | --- | --- | --- | --- | --- | --- | --- | --- | --- |
| | Intra-Event | | Inter-Event | | Overall | | Intra-Event | | Inter-Event | | Overall | |
| | EM | F1 | EM | F1 | EM | F1 | EM | F1 | EM | F1 | EM | F1 |
| *Closed-Source Models* | | | | | | | | | | | | |
| GPT-4o | 5.3 | 19.6 | 9.2 | 24.5 | 7.4 | 22.2 | 5.3 | 17.1 | 6.2 | 25.9 | 5.7 | 21.8 |
| GPT-5.2 | 10.5 | 38.0 | 12.3 | 32.6 | 11.5 | 35.1 | 19.3 | 32.9 | 7.7 | 27.4 | 13.1 | 30.0 |
| Gemini-3-Flash-Preview | 15.8 | 39.3 | 7.7 | 29.2 | 11.5 | 33.9 | 15.8 | 42.4 | 9.2 | 31.9 | 12.3 | 36.8 |
| Claude-Sonnet-4.5-20250929 | 22.8 | 44.0 | 12.3 | 35.4 | 17.2 | 39.4 | 28.1 | 48.5 | 16.9 | 39.6 | 22.1 | 43.8 |
| Gemini-3-Pro-Preview | 31.6 | 56.3 | 20.0 | 40.5 | 25.4 | 47.9 | 29.8 | 55.2 | 20.0 | 41.5 | 24.6 | 47.9 |
| Claude-Opus-4.5-20251101 | **35.1** | **57.9** | **29.2** | **53.4** | **32.0** | **55.5** | **35.1** | **60.0** | **23.1** | **50.7** | **28.7** | **55.0** |
| *Open-Source Models* | | | | | | | | | | | | |
| Qwen3-VL-235B-A22B-Thinking | 10.5 | 23.2 | 9.2 | 22.6 | 9.8 | 22.8 | 12.3 | 23.7 | 4.6 | 17.6 | 8.2 | 20.4 |
| Qwen3-VL-235B-A22B-Instruct | 12.3 | 26.1 | 6.2 | 24.6 | 9.0 | 25.3 | **17.5** | **32.1** | 6.2 | **22.9** | **11.5** | **27.2** |
| Qwen3-VL-32B-Instruct | **15.8** | 32.0 | 6.2 | 19.6 | 10.7 | 25.4 | 14.0 | 27.1 | 3.1 | 15.5 | 8.2 | 20.9 |
| GLM-4.6V | 14.0 | **34.2** | **10.8** | **27.0** | **12.3** | **30.4** | 10.5 | **32.1** | **6.2** | 20.8 | 8.2 | 26.1 |

*Table 2.* Comparison of different embedding models. Best results are highlighted in **bold**.

| Model | MAP | | | | Recall | | | | NDCG | | | |
| --- | --- | --- | --- | --- | --- | --- | --- | --- | --- | --- | --- | --- |
| | @1 | @3 | @5 | @10 | @1 | @3 | @5 | @10 | @1 | @3 | @5 | @10 |
| Qwen3-VL-Embedding-2B | **12.3** | 10.1 | 11.1 | 12.6 | 3.9 | 10.8 | 14.6 | 24.2 | **12.3** | 12.3 | 13.8 | 17.5 |
| Qwen3-VL-Embedding-8B | **12.3** | **11.0** | **12.4** | **14.1** | **4.8** | 12.5 | 17.8 | 27.0 | **12.3** | **14.1** | **16.4** | 20.1 |
| Seed-1.6-Embedding | 10.7 | 10.1 | 11.9 | 14.0 | 4.5 | **13.6** | **19.9** | **30.4** | 10.7 | 13.6 | **16.4** | **20.9** |

## 5.2. Main Results

Table 1 presents the performance of different models on DISBench. **(1) Overall Performance.** The best-performing model achieves an F1 score of 55.0 and an EM of 28.7. The low Exact Match indicates that even strong models struggle to identify complete target sets, frequently missing relevant images or including false positives. These results confirm that corpus-level contextual reasoning remains a challenging problem. **(2) Intra-Event vs Inter-Event.** We observe that stronger models exhibit a clear performance gap between the two query types, with Inter-Event queries proving substantially more difficult. In contrast, weaker models show minimal differences between the two types, likely because their limited reasoning capabilities prevent them from effectively solving either category. This pattern suggests that long-range cross-event association constitutes the primary bottleneck once basic agentic capabilities are established. **(3) Impact of Embedding Models.** Switching from the 2B to 8B embedding yields inconsistent effects, where some models improve while others degrade. This instability indicates that retrieval quality is not the systematic bottleneck, and the core challenge lies in reasoning over retrieved results.

## 5.3. Why Direct Retrieval Is Insufficient

Table 2 summarizes the performance of embedding-based retrieval on DISBench. All models achieve poor results, with Recall@3 around 10-14% and NDCG@5 only 13-17%. Moreover, even these limited scores are largely attributable to chance. Since personal photo collections contain numerous visually similar images across different events, embedding-based methods retrieve all semantically similar images indiscriminately. When some targets happen to share surface-level semantics with the query, they may be retrieved alongside distractors that violate contextual constraints. The models have no mechanism to distinguish between them. This limitation represents a fundamental ceiling of the paradigm rather than a deficiency in model capacity. To be clear, this comparison validates that DISBench queries genuinely require cross-image reasoning that the independent-matching paradigm cannot provide, rather than advocating for replacing embedding retrievers with agents in general. Stronger embeddings may retrieve visually similar images more effectively, but they cannot determine which images satisfy context-dependent constraints since the paradigm scores each image independently without access to cross-image associations. Addressing DISBench requires a fundamentally different approach capable of multi-step reasoning over corpus-level context.

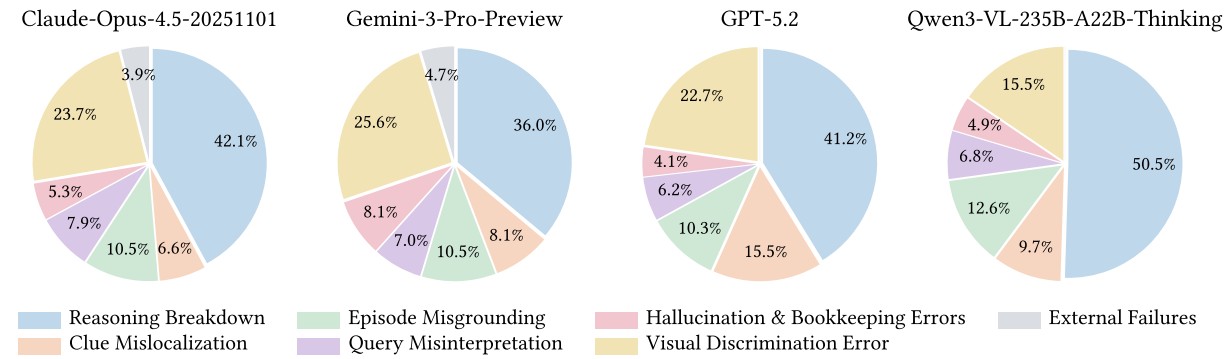

*Figure 5.* Error category distribution for four models on DISBench, identified through manual analysis of sampled failure cases. The categories span the full agent pipeline, from query understanding and context grounding to multi-step reasoning and visual perception.

*Table 3.* Ablation study of different components. We report F1 scores. The full model achieves the best performance.

| Model | Intra-event | Inter-event | Overall |
|---|---|---|---|
| **Gemini-3-Flash-Preview** | **42.4** | **31.9** | **36.8** |
| *w/o* GetMetadata Tool | 31.0 | 31.2 | 31.1 |
| *w/o* FilterMetadata Tool | 34.2 | 29.7 | 31.8 |
| *w/o* ViewPhotos Tool | 39.7 | 26.9 | 32.9 |
| *w/o* WebSearch Tool | 33.8 | 30.8 | 32.2 |
| *w/o* Explicit Memory | 34.1 | 30.1 | 31.9 |
| *w/o* Memory Compression | 38.0 | 30.4 | 33.9 |

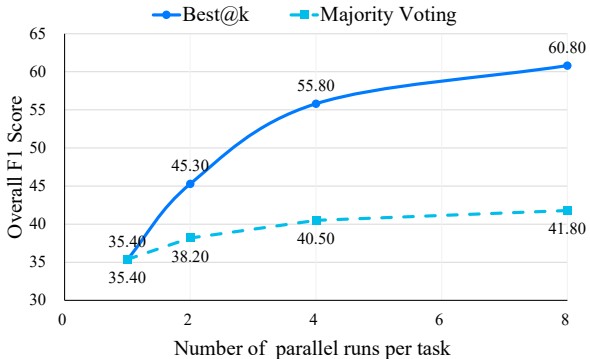

*Figure 6.* Effect of test-time scaling with different strategies. We run N parallel instances per query and aggregate via Best@k (selecting the highest-F1 result) or Majority Voting. Both strategies yield consistent improvements as N increases.

## 5.4. Ablation Study

Table 3 presents ablation results. All components contribute positively, validating our framework design. Metadata tools show the largest impact, with the removal of GetMetadata causing F1 to drop by 5.7 points. These tools enable precise temporal and spatial constraints essential for disambiguating visually similar images across events. Explicit state memory also proves critical, and its removal affects Inter-Event queries more severely than Intra-Event queries. This asymmetry aligns with task characteristics: Inter-Event queries require accumulating evidence across multiple exploration steps, and without persistent state management, agents lose track of intermediate findings. ViewPhotos and WebSearch provide complementary support for visual verification and external entity resolution, respectively. Memory compression shows the smallest impact, suggesting that the compressed representation preserves sufficient information for most queries.

## 5.5. Test-time Scaling

We investigate whether test-time scaling can improve agent performance on DISBench. Unlike deterministic retrieval, agentic exploration involves stochastic decisions that may lead to different reasoning paths. We run $N$ parallel instances per query and aggregate predictions through two strategies: Best@$k$ selects the result with the highest F1 score, and Majority Voting determines the result by majority

vote. As shown in Figure 6, both strategies yield consistent improvements as $N$ increases. In particular, the Best@$k$ metric exhibits a dramatic increase in performance, rising from 35.4 to 60.8, indicating that the model has a significant latent potential to solve the tasks. In contrast, Majority Voting significantly lags behind the Best@$k$ ceiling, suggesting that models struggle to prioritize the correct reasoning path. These findings demonstrate that while test-time scaling is a promising direction for improving agents, more robust reasoning path selection mechanisms remain a challenge.

## 5.6. Error Analysis

Figure 5 presents the distribution of error categories across four representative models, based on manual annotation of failure cases. Reasoning Breakdown emerges as the dominant error type, accounting for 36–50% of failures across all models. These errors occur when models reach the correct context but fail to execute multi-step plans, often stopping prematurely or losing track of constraints during exploration. Visual Discrimination errors constitute the second largest category, indicating that fine-grained perception, including entity identity resolution and attribute-level judgment, re-

mains a notable challenge. Episode Misgrounding and Clue Mislocalization together account for a substantial portion of errors, indicating that models struggle to anchor their search in the correct spatiotemporal context. This pattern is consistent with our earlier finding that Inter-Event queries pose greater difficulty, as they require discovering associations across distant events. These findings indicate that advancing performance on DISBench requires improvements in planning, constraint tracking, and state management rather than visual understanding alone.

## 6. Conclusion

This paper introduces DeepImageSearch, a novel paradigm that advances image retrieval from independent semantic matching to corpus-level contextual reasoning over visual histories. To support this direction, we construct DISBench through a semi-automated pipeline combining model-driven context discovery with human verification. We also designed a baseline agent framework to facilitate exploration of this task. Extensive experiments demonstrate that this task poses significant challenges for state-of-the-art models, confirming that contextual reasoning over visual histories remains an open problem. We hope this work inspires future research on agentic retrieval systems.

## Acknowledgments

This work was supported by the National Natural Science Foundation of China No. 62272467.

## Impact Statement

This paper introduces a new paradigm for image retrieval that shifts the focus from independent semantic matching to corpus-level contextual reasoning. This direction advances the field of multimodal agents by providing a challenging testbed that reveals critical capability gaps in current models. Our semi-automated data construction pipeline also offers a reusable methodology for building benchmarks that require mining complex associations at scale.

From a societal perspective, the ability to reason over personal visual histories addresses a growing need in the digital age, where people capture thousands of photos annually yet struggle to retrieve specific memories when queries are vague or context-dependent. Our work lays the foundation for intelligent assistants that help users navigate their visual memories through natural language, with potential applications in memory assistance, family archive organization, and digital legacy preservation.

We recognize that technologies operating on personal photo collections involve privacy considerations, a challenge inherent to research on personal data understanding and shared across the broader community working on personalized systems. Our benchmark is constructed from publicly licensed data following standard practices in the field, and the intended application is to assist users in searching their own collections. We believe that with appropriate safeguards in deployment, the benefits of such technology can be realized while respecting user privacy.

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

# A. Ethical Considerations

Our dataset is constructed from YFCC100M, a publicly available collection where all images were uploaded to Flickr by users who agreed to public sharing under Creative Commons licenses. We further verified that all selected photos permit research use according to their license terms. The benchmark will be released in a form that respects these licensing requirements.

We recognize that technologies for mining associations in personal photo collections could raise privacy considerations if deployed in real-world applications. However, several factors mitigate these concerns in our research context. First, all data used in this work is already publicly accessible. Second, the face clustering component serves solely to enable research on identity-based reasoning and does not involve identifying real individuals. Third, the intended application of this technology is to help users retrieve and organize their own visual memories, not to analyze others' data without consent. We encourage future work building on DISBench to consider appropriate privacy safeguards when developing systems for deployment.

# B. Limitations

**Benchmark Scale.**    Our benchmark contains 122 queries across 57 users, which is smaller in scale compared to conventional retrieval benchmarks that often contain thousands of queries. This is a common characteristic shared by benchmarks requiring complex reasoning and rigorous human verification, such as those for mathematical reasoning and agentic tasks. The 6.1% retention rate during human filtering reflects our strict quality standards rather than a deficiency in the generation pipeline. We prioritize benchmark reliability over size, as noisy annotations would undermine evaluation validity. Future work may explore more efficient annotation protocols to expand coverage while maintaining quality.

**Data Source.**    DISBench is constructed from YFCC100M, a widely adopted benchmark source in the vision-language community. While this single-source design may introduce demographic biases toward Flickr users, it ensures data consistency and reproducibility across experiments. Our semi-automated pipeline is designed to be dataset-agnostic and can be readily applied to other photo collections as they become available for research purposes.

**Metadata Availability.**    Our task formulation assumes the availability of timestamps and geographic coordinates, which may not always be present in real-world scenarios. However, this assumption aligns with the default behavior of modern smartphone cameras and mainstream cloud photo services, which automatically record such metadata unless explicitly disabled by users. Systems deployed in practice could incorporate metadata imputation techniques to handle missing values.

**Agent Framework.**    The ImageSeeker framework we propose serves as a simple baseline to facilitate future research rather than a comprehensive solution. We deliberately keep the design modular, allowing researchers to easily substitute individual components such as the retrieval backbone, memory mechanism, or planning strategy with more sophisticated alternatives. Exploring advanced techniques such as reflection, backtracking, or learned planning policies remains a promising direction for future work.

# C. Details of Automated Query Synthesis Pipeline

### C.1. Models and Parameters

**Vision-Language Models and Subgraph Sampling.**    We use Qwen3-VL-235B-A22B-Instruct for visual semantic parsing and association verification, and Gemini-3-Pro for query synthesis. For multimodal retrieval in association mining, we use Seed-1.6-Embedding as the encoder and retrieve top-5 candidates from within the source photoset and from outside, respectively. Sampling terminates at 40 edges.

**Face Processing.**    We utilize the buffalo_l model pack from InsightFace for face detection and recognition, with input images resized to $640 \times 640$ pixels. To ensure high precision, we apply a denoising strategy that filters out detections with a confidence score below 0.68. The remaining face embeddings are grouped via agglomerative clustering, utilizing a cosine distance threshold of 0.65. To further mitigate noise from spurious detections, we discard any clusters containing fewer than three faces.

---

**Algorithm 1** Balanced Subgraph Sampling

---

1: **Input:** Memory graph $\mathcal{G}$, pivot photo $p$, edge limit $L$
2: **Output:** Sampled subgraph $\mathcal{S}$
3: $\mathcal{S}$.nodes $\leftarrow \{p\}, \mathcal{S}$.edges $\leftarrow \emptyset$
4: $\mathcal{F} \leftarrow \{p\}$
5: **while** $|\mathcal{S}$.edges$| < L$ **and** $\mathcal{F} \neq \emptyset$ **do**
6:    $u \leftarrow$ UNIFORMSAMPLE($\mathcal{F}$)
7:    $\mathcal{N} \leftarrow \{v \in$ NEIGHBORS$(u, \mathcal{G}) \mid (u, v) \notin \mathcal{S}$.edges$\}$
8:    **if** $\mathcal{N} = \emptyset$ **then**
9:       $\mathcal{F} \leftarrow \mathcal{F} \setminus \{u\}$
10:       **continue**
11:    **end if**
12:    *// Balanced edge-type sampling*
13:    $\mathcal{T} \leftarrow$ group $\mathcal{N}$ by EDGETYPE$(u, \cdot)$
14:    $t^* \leftarrow$ UNIFORMSAMPLE($\mathcal{T}$.keys)
15:    $v \leftarrow$ UNIFORMSAMPLE($\mathcal{T}[t^*]$)
16:    $\mathcal{S}$.edges $\leftarrow \mathcal{S}$.edges $\cup \{(u, v)\}$
17:    **if** $v \notin \mathcal{S}$.nodes **then**
18:       $\mathcal{S}$.nodes $\leftarrow \mathcal{S}$.nodes $\cup \{v\}$
19:       $\mathcal{F} \leftarrow \mathcal{F} \cup \{v\}$
20:    **end if**
21: **end while**
22: *// Context completion*
23: **for all** VisualEntity node $e \in \mathcal{S}$ without parent Photo **do**
24:    Add parent Photo node and edge to $\mathcal{S}$
25: **end for**
26: **for all** Photo node $p \in \mathcal{S}$ without parent Photoset **do**
27:    Add parent Photoset node and edge to $\mathcal{S}$
28: **end for**
29: **return** $\mathcal{S}$

---

### C.2. Subgraph Sampling Algorithm

Algorithm 1 presents our sampling procedure. We maintain a frontier set of expandable nodes. At each iteration, we randomly select a node from the frontier, then apply balanced edge-type sampling: group its unexpanded edges by type, uniformly sample an edge type, then uniformly sample an edge of that type. If a node has no unexpanded edges, it is removed from the frontier. This design prevents structural edges (which are dense within events) from dominating and ensures cross-event association edges are adequately explored. After reaching the edge limit, we run a context completion step that adds parent Photo nodes for orphan VisualEntity nodes and parent Photoset nodes for orphan Photo nodes.

### C.3. Quality Control Criteria

**Association Verification.** We employ a binary decision system to determine whether images from different photosets contain identical visual elements. An association is confirmed when the model verifies that the target image contains the same element as specified in the query image clue. Confirmation requires satisfaction of one of the following criteria. First, unique identifiers must be present, including license plates, serial numbers, or distinctive defects. Second, highly matched visual features must be combined with supporting metadata, such as similar items within private spaces or identical locations photographed at different times. Third, clear reference relationships must exist where the query image content appears within the target image, for example, on screens, posters, or picture frames. Fourth, different manifestations of the same theme must be evident, such as a physical object alongside its promotional poster. An association is rejected when entities exhibit clearly different features, belong to the same category but represent different individuals, or show no relation whatsoever. Only confirmed matches are retained as association edges in the memory graph.

*Table 4.* **Tool interface specifications.** For each tool, we list its functionality, input parameters, and output format. Optional parameters are marked with [†].

| Tool | Functionality | Input Parameters | Output |
|---|---|---|---|
| ImageSearch | Multimodal similarity search over the photo collection. | `text`: text query describing target content[†]
`photos`: photo IDs as visual query cues[†]
`top_k`: number of results (default: 20)
`save_as`: subset name to save results[†]
`search_within`: subset name to restrict scope[†] | List of (photo_id, score) tuples ranked by similarity. |
| GetMetadata | Retrieve structured metadata for specified photos. | `photos`: list of photo IDs
`fields`: subset of fields to return (available values: [time, address])[†] | Per-photo metadata including `time` (ISO format) and `address`. |
| FilterMetadata | Filter photos by metadata constraints using boolean expressions. | `expression`: Python boolean expression over `time` and `address` variables
`save_as`: subset name to save results[†]
`filter_within`: subset name to restrict scope[†] | Count and list of photo IDs satisfying the expression. |
| ViewPhotos | Inject photos into the agent's visual context for direct inspection. | `photos`: list of photo IDs (max 20) | Visual artifact for agent perception. |
| WebSearch | External web search to resolve entities in user queries. | `query`: search query string
`top_k`: number of results | List of results with URL, title, and extracted text. |

**Query Synthesis Constraints.** We enforce three requirements for query synthesis. The first requirement is visual ambiguity. Targets must possess a low-distinctiveness appearance, and the corpus must contain visually similar distractors that cannot be distinguished through appearance alone. The second requirement is contextual identifiability. Target uniqueness must derive from associations with events, persons, or temporal context rather than from visual features. The third requirement is a strong-to-weak reasoning flow. Queries should utilize easily retrievable anchor images with distinctive features to locate targets with weak visual characteristics. These constraints are verified by examining whether the removal of contextual clues from the query would render the target indistinguishable from distractors.

### C.4. User Selection Criteria

To ensure data quality and diversity, we apply a two-stage user selection pipeline. In the first stage, we filter YFCC100M users based on objective criteria. Each user must have at least 2,000 photos with a metadata coverage rate of at least 90%, a minimum of 50 photosets, and an account history spanning more than one year. In the second stage, we randomly sample 100 users from the qualified pool while ensuring geographic diversity across continents except Antarctica. After the query synthesis and human verification process, 57 users retain at least one valid query and are included in the final benchmark. For each selected user, we accumulate complete photosets chronologically until reaching a capacity of 2,000 photos, resulting in an average temporal span of 3.4 years per user as reported in Section 3.

## D. Agent Implementation Details

### D.1. Tool Specifications

Several design choices merit further explanation. First, the `photos` parameter in ImageSearch serves as visual query cues rather than defining the search scope. This allows agents to use previously discovered photos as reference examples while still searching across the entire collection or a specified subset. Second, both ImageSearch and FilterMetadata support explicit state management through `save_as` and `search_within`/`filter_within` parameters, enabling agents to utilize previous tool results and incrementally narrow down candidates across multiple reasoning steps.

For FilterMetadata, we implement a `match_address(address, query)` function to handle address matching robustly. This function first normalizes common aliases (e.g., "US" → "United States") using a predefined dictionary, then checks whether the normalized query appears as a substring in the metadata address. When initial normalization fails to match

any address in the collection, the function falls back to a geocoding service[1] for resolution. This design accommodates the variability in how locations are expressed in both user queries and photo metadata. For WebSearch, we leverage the Serper API[2] to retrieve the top-$k$ results, aggregating the rank, title, and snippet into the final search output.

### D.2. Memory Implementation

**Explicit State Memory.** Photo subsets are maintained as a dictionary mapping subset names to lists of photo IDs. This structure is stored in the agent state and persists throughout the session. Agents create subsets through the `save_as` parameter when calling ImageSearch or FilterMetadata, and reference existing subsets via `search_within` or `filter_within` parameters. This mechanism enables multi-step reasoning patterns such as: (1) filtering photos by time range and saving as `trip_photos`, (2) searching within `trip_photos` for specific content, and (3) further filtering by location constraints.

**Compressed Context Memory.** When the conversation history approaches the context length limit (128K tokens in our experiments), the system compresses previous interactions into a structured summary. This summary consists of two components: *session memory* captures high-level goals and key findings accumulated throughout the session, while *working memory* records the current subgoal and immediate action plans. The compression is triggered either automatically when the token count exceeds a threshold, or manually when the agent invokes a dedicated compression tool. After compression, the full interaction history is replaced by the structured summary, freeing context space for continued exploration while preserving essential reasoning state.

### D.3. System Prompt

The agent's planning and reasoning behavior is guided by a structured system prompt. We highlight the key design elements below.

**Query Understanding Framework.** To ensure systematic query analysis, the prompt instructs the agent to decompose each query into three components:

- *Episode*: The latent spatiotemporal context implied by the query, representing a coherent event or sequence of events.
- *Episode Breakdown*: A step-by-step logical path that decomposes the episode into individual photos or sub-events, with explicit relations (e.g., same time, same location, same person) connecting them.
- *Target*: The specific photos to be returned, described along with visual content and metadata constraints.

This decomposition separates context inference from target identification, preventing the agent from conflating anchor constraints with target requirements.

**Key Instructions.** The prompt emphasizes several behavioral guidelines: (1) Anchor photos that help locate events should not impose their visual features on target photos unless explicitly stated. (2) Temporal phrases like "on the day we visited..." primarily constrain time or location rather than requiring the referenced event to appear in results. (3) The agent must operate autonomously without requesting user clarification—when information is ambiguous, it should make best-effort inferences and proceed. (4) All responses must conclude with an explicit answer in the format: `"The final answer is: [photo_id1, photo_id2, ...]."` to ensure unambiguous output parsing.

### D.4. Experimental Configurations

**Backbone Models.** For agentic evaluation, we access proprietary models through their official APIs. Open-source models are served locally using vLLM with the OpenAI-compatible API interface. All models use default temperature settings and are equipped with identical tool interfaces.

**Embedding Models.** For the ImageSearch tool, we use Qwen3-VL-Embedding as the multimodal encoder. We evaluate two model sizes (2B and 8B parameters) to assess the impact of retrieval quality on overall agent performance. The embedding index is built per user, containing all photos in their visual history. For direct retrieval baselines, we additionally

---

[1] https://opencagedata.com/
[2] https://serpapi.com/

evaluate Seed-1.6-Embedding to cover both open-source and proprietary embedding models.

**Hyperparameters.** We set the maximum number of interaction turns to 30, which we found sufficient for most queries while preventing infinite loops. The context length limit is set to 128K tokens, after which compressed context memory is triggered. For retrieval operations, ImageSearch returns 20 results by default, and ViewPhotos accepts at most 20 photo IDs per call to balance visual inspection coverage with context consumption. Context compression is performed by GPT-4o-mini. All backbone models use their default temperature settings. The embedding model for multimodal retrieval is Qwen3-VL-Embedding (2B and 8B variants as specified in experiments).

**Evaluation Protocol.** Each query is executed independently with a fresh agent state. The agent interacts with tools until it produces a final answer or reaches the maximum turn limit (30 turns). We extract predicted photo IDs from the agent's final response and compute set-level Exact Match (requiring identical sets) and F1 score (harmonic mean of precision and recall) against ground-truth annotations. For retrieval baselines, we report MAP@$k$, Recall@$k$ and NDCG@$k$ with $k \in \{1, 3, 5, 10\}$.

## E. Qualitative Examples

To provide intuition for how agents navigate visual histories, we present two representative traces: a successful case demonstrating effective multi-event disambiguation, and a failure case illustrating common error patterns.

---

**Case 1: Successful Multi-Event Disambiguation**

**Query:** *Find all photos with the sea taken at the beach two days after watching the fireworks show.*

**Step 1: Anchor Event Discovery**

`ImageSearch("fireworks show")` → 15 candidates

Visual inspection reveals fireworks photos from multiple distinct events.

**Step 2: Temporal Grounding**     Key Insight

`GetMetadata([15 photos])` → Three fireworks events identified:

    2012-08-03 (Bournemouth) — 2012-06-04 (Wendover) — 2011-07-29 (Bournemouth Pier)

Agent recognizes temporal ambiguity and plans to check all candidate dates.

**Step 3: Parallel Date Filtering**

`FilterMetadata(date="2012-08-05")` → 26 photos → `aug_5`

`FilterMetadata(date="2012-06-06")` → 0 photos             (branch pruned)

`FilterMetadata(date="2011-07-31")` → 8 photos → `jul_31`

**Step 4: Scoped Target Search**

`ImageSearch("sea beach", within=jul_31)` → 3 high-confidence matches

`ImageSearch("sea beach", within=aug_5)` → low similarity scores

**Step 5: Visual Verification**

`ViewPhotos([jul_31 candidates])` → 3/8 confirmed with sea visible

`ViewPhotos([aug_5 top results])` → 0/20 match (all parade/carnival photos)

Agent correctly determines Aug 5 was spent at town events, not at beach.

**Output:** `[6009707544, 6009157655, 6009152901]`       ✓ Correct

---

**Case 2: Visual Discrimination Failure**

**Query:** *Find close-up photos of the spire-topped building that appears in the distance in the first photo taken on April 18, 2014.*

**Step 1: Anchor Photo Identification**

`FilterMetadata(date="2014-04-18")` → 27 photos

`GetMetadata([27 photos], fields=[time])` → earliest is `13934079356` at 10:09

**Step 2: Anchor Inspection**

`ViewPhotos([13934079356])`

Agent identifies: rural landscape with a church spire visible in a distant village.

**Step 3: Target Search**

`ImageSearch("close-up church tall spire")` → 30 candidates
`ImageSearch([anchor], "church spire close up")` → 30 candidates

**Step 4: Candidate Inspection**                                          Error Point

`ViewPhotos([30 candidates])` — Agent reviews results including:

   `13934133936` (rank 1, score=0.94): labeled ✓ "*same church*"

   `13933619246` (rank 12, score=0.63): labeled ✗ "*different church*"         Missed GT

Agent fails to recognize that `13933619246` depicts the same building from a different angle.

**Step 5: Verification (Incomplete)**

`ViewPhotos([13934079356, 13934133936])` → confirms match

Agent terminates without re-examining other candidates.

**Output:** `[13934133936]`                                          ✗ Incorrect

**Ground Truth:** `[13956729065, 13933619246]`

- - - - - - - - - - - - - - - - - - - - - - - - - - - - - - - - - - - - - - - - - - - - - - - - - - -

**Error Analysis:**

- **Visual Discrimination:** `13933619246` was retrieved (rank 12) but incorrectly dismissed as a "different church" during visual inspection, despite depicting the same building.

- **Retrieval Gap:** `13956729065` never appeared in any search results, indicating insufficient query diversity or embedding coverage.

- **Missing Location Prior:** Because the target building is very small in the anchor, pure embedding retrieval is high-noise. The agent did not apply a location-based pre-filter around the anchor, leading to an unnecessarily large and noisy candidate set.

