# OpenReview forum: "DeepImageSearch: Benchmarking Multimodal Agents for Context-Aware Image Retrieval in Visual Histories"
_ICML.cc/2026/Conference — ICML 2026 regular_

### Official Review · Reviewer_b2G4 · 2026-03-11

**Soundness:** 4
**Presentation:** 4
**Significance:** 3
**Originality:** 4
**Overall Recommendation:** 5
**Confidence:** 4

**Summary:**

The paper introduces DeepImageSearch, a challenging image retrieval setting that is difficult for standard single-shot retrieval models. The task requires corpus-level contextual reasoning over a sequence of visual histories to identify the correct target. To support evaluation, the authors present DISBench, a benchmark constructed via a human-model collaborative pipeline intended to ensure both sufficient reasoning depth and high data quality. The paper also proposes an agent-based baseline, and reports extensive experiments across both proprietary and open-source LLMs.

**Compliance With Llm Reviewing Policy:**

Affirmed.

**Ethical Review Concerns:**

Updated: Not a concern anymore

**Final Justification:**

The rebuttal addresses all of my concerns. However, when reading other reviewers, there are some small weaknesses for this work including more analyses on the method/ model size and the benchmark scale. I updated my reviews to remove my ethical review concerns and keep my initial score.

**Key Questions For Authors:**

1. In ImageSeeker, how sensitive is end-to-end performance to the quality of the embedding model used for candidate retrieval? Since ImageSearch uses the Qwen-Embedding model to retrieve candidates, is the final performance effectively bounded by the recall of Qwen-Embedding at the candidate set size?

2. The paper’s example queries appear quite complex. Do you have evidence on how common DeepImageSearch-style queries are in real-world retrieval settings? For instance, what is the estimated frequency of such multi-step, context-dependent queries in practical systems, and in which application domains do you expect them to matter most?

3. What is the human performance on this task when given the same tools/interface, and how long do humans typically take per query? Reporting human accuracy and completion time would help contextualize task difficulty and evaluate the practical applicability of the benchmark.

4. What is the per-query complexity of the proposed agent baseline in terms of latency and cost, and how does this trade off against accuracy? A clear performance-efficiency analysis would be important for assessing feasibility at scale.

**Limitations:**

The paper should more explicitly discuss the limitations of the proposed task and framework. In particular, it would be helpful to address pipeline efficiency and scalability in real-world information retrieval systems, and to clarify the practical constraints these factors impose on deployment.

**Strengths And Weaknesses:**

- Soundness: The work provides substantial detail on dataset construction, framework implementation, and experimental setup. The evaluation spans multiple model settings and includes recent leading models with strong reasoning capability.
- Presentation: The figures and tables are well-designed, and the paper is generally well-written and easy to follow. One area for improvement is the captions, which could be expanded to be more self-contained and interpretable without repeatedly referring back to the main text.
- Significance: The proposed task could open new research directions for advanced information retrieval systems, and the emphasis on planning and reasoning may help narrow the gap between system performance and human performance. My main concern is efficiency and scalability when deploying such approaches in large-scale systems (e.g., with billions of documents). In addition, the task may capture relatively rare or edge-case scenarios, and the paper does not yet provide evidence that such scenarios will be broadly prevalent in future real-world applications.
- Originality: The contributions, new task formulation, comprehensive benchmark construction, and extensive experimentation, appear valid and novel. In particular, the human-model collaborative pipeline is a thoughtful design choice that helps support dataset quality.

---

> ### Author Rebuttal · Authors · 2026-03-31
>
> We thank the reviewer for the thorough evaluation. We conducted a human study and per-query cost analysis during rebuttal to address the four questions.
>
> **Presentation:** We have expanded all captions in the revision to be self-contained.
>
> ---
>
> **Q1: Embedding Sensitivity and Recall Upper Bound**
>
> The embedding model is not the effective bottleneck in our pipeline, for three reasons.
>
> **(1) The agent does not retrieve with the raw query in a single shot.** It decomposes complex queries into sub-queries (e.g., "beach photos two days after fireworks" first becomes "fireworks show" to locate the event, then "sea beach" after date inference). Each sub-query aligns well with its targets, so Table 2's low recall on full queries does not reflect actual pipeline constraints. **(2) At the sub-query level, current SOTA embeddings are adequate.** Qwen3-VL-Embedding achieves over 80% Recall@1 on MMEB, and ImageSearch returns top-20 per call. **(3) Table 1's embedding swap confirms this empirically.** Claude-Opus achieves nearly identical F1 under 2B and 8B embeddings (gap only 0.5), while backbone gaps under the same embedding exceed 33 F1 points. The bottleneck lies in query decomposition, planning, and inference.
>
> ---
>
> **Q2: Query Frequency and Application Scenarios**
>
> **(1) Context-dependent queries reflect natural retrieval behavior.** Users searching personal photos rely on contextual cues rather than visual descriptions, and existing tools poorly support this (Naaman et al., MM 2004; Whittaker et al., PUC 2010). As photo volumes grow and remembering dates becomes harder, such queries will only be more prevalent.
>
> **(2) Demand is suppressed by current systems.** When systems only offer keyword matching, users learn to issue simple queries. This does not mean complex needs are absent, just as users typed short keywords before LLMs not for lack of complex needs.
>
> **(3) Applications include** personal memory assistance (especially elderly and cognitively impaired users), family archives, digital legacy preservation, and forensic timeline reconstruction.
>
> **(4) The benchmark targets the capability ceiling.** Just as MATH tests abilities beyond everyday homework, DISBench evaluates a **superset** of simple retrieval abilities. Added in revision.
>
> ---
>
> **Q3: Human Performance**
>
> Since human annotators must operate the full tool interface step by step, per-query evaluation is time-intensive. We sampled 10 queries (5 intra + 5 inter, stratified), with annotators using the same tools.
>
> | | EM | F1 |
> |---|---|---|
> | Human | 0.70 | 0.77 |
> | Claude-Opus-4.5 (same queries) | 0.50 | 0.63 |
>
> **The gap concentrates on Inter-Event queries.**
>
> | Split | Human F1 | Agent F1 |
> |---|---|---|
> | Intra-event | 0.80 | 0.97 |
> | Inter-event | 0.73 | **0.29** |
>
> On Inter-Event queries human F1 is 2.5x the agent's, consistent with Section 5.2. This confirms that DISBench targets a meaningful capability gap rather than an artificial ceiling.
>
> **(1) Humans use fundamentally different strategies.** Humans need only **5.5 tool calls** per query while the agent needs **21.3**, yet achieve higher F1. 41.8% of human operations are metadata filters compared to only 9.9% for the agent. This gap points to a concrete direction for improving agent planning.
>
> **(2) Completion time is not meaningful at this sample size.** We plan to report timing after expanding the study.
>
> ---
>
> **Q4: Per-Query Cost and Scalability**
>
> DISBench targets personal collections (thousands to tens of thousands of photos), not web-scale retrieval. Per-query cost (8B retriever, all queries averaged):
>
> | Model | Turns | Tool Calls | Tokens (K) | Cost (USD) |
> |---|---|---|---|---|
> | Claude-Opus-4.5 | 16.8 | 22.1 | 315 | 5.01 |
> | Gemini-3-Pro | 15.8 | 19.6 | 427 | 0.93 |
> | Gemini-3-Flash | 20.3 | 20.7 | 480 | 0.08 |
>
> **(1) Low-cost deployment is already viable.** Flash costs 0.08 USD/query while Opus at 5.01 USD achieves the best accuracy. As reasoning improves, agents should complete tasks in fewer turns, further reducing cost.
>
> **(2) Complexity depends on query and corpus characteristics, not collection size.** The agent retrieves via an embedding model, so per-call computation is independent of scale. Reasoning turns should remain similar in larger settings. Added in Limitations.
>
> ---
>
> **Ethics: Privacy/PII**
>
> We have implemented multiple safeguards.
>
> **(1) Limited metadata only.** Only timestamps and GPS (already public) are used. No social attributes are included.
>
> **(2) Strict de-identification.** Face clustering yields only anonymized Person IDs. Embeddings and clusters are not released. Only Flickr photo IDs and annotations are distributed, not original images. GPS reduced to ~1km.
>
> **(3) Access control.** A Data Use Agreement prohibiting re-identification is required, with requests reviewed by authors. Details in Appendix A. DISBench is built from YFCC100M, widely used at CVPR/ICCV/ECCV, following standard practice.
>
> ---
>
> We thank the reviewer and welcome further discussion.

---

> > ### Author Rebuttal · Reviewer_b2G4 · 2026-04-01
> >
> > Thanks for the rebuttal. It addresses my concerns but the limitations and significance are still considerable issues. I keep my initial score.

---

> > > ### Author Response · Authors · 2026-04-05
> > >
> > > We thank the reviewer for the thorough evaluation and support. The recognition of our problem formulation, benchmark construction pipeline, and experimental comprehensiveness is very encouraging. The discussions on human performance, per-query cost, and application scenarios have also helped us improve the paper.
> > >
> > > Regarding the significance and limitations the reviewer mentioned, we understand these are inherent open challenges for this new direction and briefly elaborate here.
> > >
> > > (1) On significance, as personal photo collections continue to grow and users increasingly need precise memory retrieval, the practical demand for such context-aware queries will become more prominent. Meanwhile, as the reasoning capabilities of frontier models continue to advance, complex retrieval tasks that were previously infeasible will become achievable, further unlocking user demand in this area.
> > >
> > > (2) On scalability, DISBench targets personal photo collections, where retrieval is performed via embedding models and per-call computation is independent of collection size, making practical deployment viable.
> > >
> > > We will further strengthen the discussion of both aspects in the camera-ready. The reviewer's constructive feedback has been very helpful in improving this work, and we are grateful for the engagement.

---

### Official Review · Reviewer_sLBf · 2026-03-12

**Soundness:** 2
**Presentation:** 3
**Significance:** 2
**Originality:** 3
**Overall Recommendation:** 5
**Confidence:** 2

**Summary:**

The submission frames a new challenging image search problem to retrieve images according to textual or multimodal queries from a spatiotemporally correlated candidate set, i.e., visual histories. While existing image search solutions, either embedding-based or reasoning-based, take candidate images as independent items to be matched with the query, the key question studied here is how to explore the intrinsic correlations among images for better matching with vague queries accurately. To answer this question, the authors built a new benchmark (DISBench) composed of queries and their corresponding images that can only be retrieved with a good understanding of the spatiotemporal correlations among image candidates. Agentic baselines were built and tested to demonstrate the challenges of the task and the limitations of existing image search solutions.

**Compliance With Llm Reviewing Policy:**

Affirmed.

**Final Justification:**

The follow-up clarification provided by the authors addresses my concerns about comparison fairness. I'm happy to vote for accept.

**Key Questions For Authors:**

1. How will the reasoning-based or embedding-based solutions perform against the agentic baselines if using models at similar sizes?
2. Are those metadata provided to Qwen3-VL-embedding as well? If not, will it be helpful to encode them together with the images into embeddings for implicit correlating candidates and matching them with the queries?
3. How many API calls and how much time are needed for searching given one query on average? What about the annotation cost for the possibility of scaling up the benchmark?

**Limitations:**

Yes

**Strengths And Weaknesses:**

The paper is technically sound in general. The semi-automatic data annotation pipeline and the agentic baselines are designed reasonably. However, the experiment can be improved by a fairer and more direct comparison between existing embedding-based and reasoning-based solutions with the agentic baselines. Even though the authors have reported results of two embedding-based solutions separately from the agentic baselines, the metrics are different, which makes it less intuitive for understanding how large the performance gap between the two types of solutions is. Moreover, the agentic baselines make use of large-scale VLM (e.g., Qwen3-VL-235B-A22B), but the embedding-based solutions are much smaller (e.g., Qwen3-VL-Embedding-8B). This makes the comparisons less fair. In terms of the benchmark, its limited size with only 122 queries raises the concern about how universal the conclusions made in this paper are.

While the image search task studied in this paper is intuitive, its importance in practice is unclear. The provided examples appear to be artificially constructed with highly ambiguous queries. More qualitative studies will be helpful.

The manuscript is well-structured and easy to follow. The limitations of existing image search solutions on the proposed challenging setup are discussed. The task studied here is new and challenging to existing image search solutions.

---

> ### Author Rebuttal · Authors · 2026-03-31
>
> We thank the reviewer for the constructive feedback. We address each concern below.
>
> **W1: Fairness of Embedding vs. Agent Comparison**
>
> **(1) We designed the comparison to give embeddings their best chance.** Embeddings produce ranked lists while agents produce sets, so we used the most favorable metric for each. Our core argument is not "agents beat embeddings," but that per-image independent scoring cannot solve queries requiring cross-image reasoning. The unified metric experiment in Q1 provides direct evidence under the same EM/F1 metrics.
>
> **(2) The bottleneck is at the paradigm level, not model capacity.** The 8B Qwen3-VL-Embedding is retrieval-specialized and already at SOTA on mainstream benchmarks. For "photos of the lead singer at the concert with a blue-and-white logo," the logo appears in a *different* image from the targets. The needed information lies outside the image being scored, and no embedding, regardless of size, can access this signal.
>
> ---
>
> **Q1: Unified Metric Experiment**
>
> We granted embeddings an **oracle advantage unavailable in practice**: the ground-truth target count k per query (agents do not receive this). We computed EM/F1 on top-k results.
>
> | Method | EM | F1 |
> |---|---|---|
> | Qwen3-VL-Emb-2B | 1.6 | 11.3 |
> | Qwen3-VL-Emb-8B | 2.5 | 12.6 |
> | Seed-1.6-Emb | 2.5 | 10.6 |
>
> Even under this unrealistic upper bound, the best F1 is only 12.6. The 2B-to-8B gain is only 1.3 F1 points, confirming that scaling cannot break the paradigm bottleneck.
>
> ---
>
> **Q2: Metadata-Augmented Embedding**
>
> We encoded timestamps and GPS as text jointly with images. Per-query analysis:
>
> | Direction | Queries | Proportion |
> |---|---|---|
> | Improved | 21 | 17% |
> | Degraded | 14 | 12% |
> | Unchanged | 87 | 71% |
>
> 71% unchanged directly confirms our argument. Metadata enriches single-image representations but does not change independent scoring. "Beach photos two days after fireworks" requires first locating the fireworks event then computing the date. This cross-image reasoning cannot be replaced by enriching individual images.
>
> ---
>
> **W2: Practical Importance**
>
> **(1) The complexity of DISBench queries reflects a genuine gap in current systems.** We agree that these queries look different from traditional retrieval, and that is exactly the gap we aim to formalize. If queries were specific enough for direct matching, the task would reduce to conventional retrieval with no need for a new paradigm.
>
> **(2) This gap has solid empirical grounding.** Users searching personal photos rely on contextual cues rather than visual descriptions, and existing tools poorly support this (Naaman et al., MM 2004; Whittaker et al., PUC 2010). Apple Photos' Memories already mines cross-event contextual associations in production, reflecting real industry demand for this capability. DISBench formalizes it into a quantitatively evaluable benchmark.
>
> **(3) DISBench's value extends beyond retrieval.** It reveals that reasoning breakdown accounts for 36–50% of errors, offering insights for planning and reasoning research. More qualitative examples will be added.
>
> ---
>
> **W3: Benchmark Scale**
>
> **(1) The current scale reflects a deliberate quality-quantity tradeoff.** Retained queries must contain multiple visually similar distractors resolvable only through cross-image reasoning. Models solving these hard queries can inherently solve easier ones, ensuring discriminative power. This scale is comparable to similar multimodal deep-search benchmarks with rigorous human verification (MM-BrowseComp 224, MMSearch 300, MMDR-Bench 140).
>
> **(2) Despite the moderate size, model rankings are statistically stable.** Split-half reliability over 10,000 splits yields mean Spearman ρ=0.911. Bootstrap 95% CIs show the top-3 set unchanged in 97.5% of resamples (Claude-Opus 55.0[47.7,62.2], Gemini-Pro 47.9[40.5,55.2], Claude-Sonnet 43.8[36.7,51.2]).
>
> **(3) Expansion is underway along two directions.** First, 94 new queries at a relaxed difficulty threshold have completed Stages 1–3 of the pipeline, with only human review remaining. Second, we are onboarding users from more diverse demographic and geographic backgrounds.
>
> ---
>
> **Q3: Inference Efficiency and Annotation Cost**
>
> **(1) Low-cost deployment is already viable.** Gemini-3-Flash costs only 0.08 USD/query. Claude-Opus-4.5 costs 5.01 USD/query but achieves the best accuracy.
>
> **(2) More turns correlate with stronger performance, not inefficiency.** Top-3 models average **16.7 turns** while bottom-3 average only **7.4 turns**. Weaker models terminate prematurely due to insufficient planning. This aligns with Section 5.5.
>
> **(3) Scaling up the benchmark is cost-effective.** Pipeline VLM API costs approximately 2,600 USD for 109K images. For the 94 easy/dev queries mentioned above, Stages 1–3 of the automated pipeline are already complete, so only human review remains.
>
> ---
>
> We thank the reviewer again and welcome further discussion.

---

> > ### Author Rebuttal · Reviewer_sLBf · 2026-04-01
> >
> > Most of my concerns have been resolved but there are still a couple of questions regarding the comparison with the embedding-based solutions:
> >
> > 1. For the inconsistent model sizes, if the authors believe in scaling up embedding-based retrieval solutions won't help, is it possible to scale down the agent-based solution? It is a practical problem especially when being deploy to devices with limited capacity.
> >
> > 2. For fair comparison, if the reason of not using the same metrics is the different types of retrieved results (ranked list vs set), can the agent return a sorted list instead, to be evaluated using the metrics adopted by embedding-based method?

---

> > > ### Author Response · Authors · 2026-04-02
> > >
> > > We thank the Reviewer for carefully reading our first-round response and for these two insightful follow-up questions. Before presenting new experiments, we want to candidly clarify a point that we may not have communicated well previously, which we believe is the root cause of the Reviewer's continued focus on this issue.
> > >
> > > **On the role of the embedding comparison in our paper.** We did not include embedding baselines to argue that "agents are better than embeddings." A system that can interact over multiple turns, call tools, and inspect images outperforming single-pass vector matching on complex reasoning tasks is not surprising, and we do not consider this a meaningful scientific finding. The real purpose is to serve as a **validity check for DISBench's task design**: if embeddings could achieve reasonable scores, our queries would fail to capture cross-image reasoning, and the benchmark design would be flawed. The low embedding scores validate the benchmark, not the relative merits of agents versus embeddings.
> > >
> > > Due to space constraints we did not elaborate on this in Round 1, which may have made our experiments read as "proving agents win under different conditions." That was not our intent. Both experiments below should be understood in this context: they rule out confounding variables and confirm that DISBench measures a capability genuinely beyond the reach of independent scoring.
> > >
> > > ---
> > >
> > > **Q1: Scaling Down the Agent**
> > >
> > > This question targets a key confound: agents used models up to 235B while embeddings used only 8B. How much of the gap comes from parameter disparity? The cleanest design is what the Reviewer suggests: scale the agent down to match.
> > >
> > > We ran Qwen3-VL-8B-Instruct as the agent, which has the same parameter count as Qwen3-VL-Embedding-8B.
> > >
> > > | Method | Params | Oracle k | Total F1 |
> > > |---|---|---|---|
> > > | Qwen3-VL-Embedding-8B | 8B | Yes | 12.6 |
> > > | Qwen3-VL-8B-Instruct (agent) | 8B | No | 20.5 |
> > >
> > > **(1) The gap remains significant after fully controlling for parameter count.** Both models have the same parameters, and the embedding even enjoys an oracle k advantage (knowing the ground-truth target count, which the agent does not). The agent still leads by 63%. With parameters controlled, this gap can only be attributed to whether the method performs cross-image reasoning.
> > >
> > > **(2) This gap stems from a structural property of the task, not insufficient model capacity.** Consider the paper's example: "Find photos from the musical performance identified by the blue and white event logo on site, where only the lead singer appears on stage." The collection contains multiple visually similar concerts. The only distinguishing clue is a blue-and-white logo appearing in a different image, not in any target photo. An embedding model scoring each image independently cannot access the logo image, so it cannot determine which concert is correct. This blind spot is structural and cannot be overcome by scaling up embeddings.
> > >
> > > **(3) 8B agent is a viable lightweight deployment option.** An 8B model already suffices for basic tool-calling and cross-image reasoning, making it suitable for on-device deployment.
> > >
> > > ---
> > >
> > > **Q2: Unified Retrieval Metrics**
> > >
> > > This question targets another confound: could the conclusion depend on metric choice? We converted agent predictions into ranked lists (by output order), padding with incorrect items when predictions fall short of K, and computed the same metrics used for embeddings:
> > >
> > > | Method | Type | MAP@5 | MAP@10 | Recall@10 | NDCG@10 |
> > > |---|---|---|---|---|---|
> > > | Qwen3-VL-Emb-8B | embed | 12.2 | 14.0 | 26.3 | 19.8 |
> > > | Seed-1.6-Emb | embed | 11.9 | 14.0 | 30.4 | 20.9 |
> > > | Claude-Opus-4.5 | agent | 52.7 | 54.6 | 58.4 | 59.5 |
> > > | Gemini-3-Pro | agent | 47.5 | 51.4 | 51.4 | 51.4 |
> > >
> > > **(1) The conclusion is robust to metric choice.** Whether measured by set-based EM/F1 or unified retrieval metrics, the gap magnitude is highly consistent, ruling out metric selection as a confound. The embedding models also show minimal internal differences, corroborating Q1.
> > >
> > > **(2) The Recall vs MAP gap reveals "coincidental hits."** Seed-1.6-Emb reaches Recall@10 of 30.4 but MAP@10 of only 14.0. Embeddings do retrieve some targets, but these hits are essentially coincidental. Personal photo collections contain many visually similar images across events, and embeddings return all of them indiscriminately. Some targets happen to be included, but the model cannot distinguish which satisfy the contextual constraints and which do not.
> > >
> > > ---
> > >
> > > We will integrate both experiments into the paper and rewrite Section 5.3 to clearly frame the embedding comparison as a benchmark validity check. We hope this round has fully addressed the Reviewer's concerns, and we are grateful for the probing questions across both rounds that have substantively improved our paper's rigor and clarity.

---

### Official Review · Reviewer_Ny6X · 2026-03-13

**Soundness:** 3
**Presentation:** 3
**Significance:** 2
**Originality:** 2
**Overall Recommendation:** 4
**Confidence:** 4

**Summary:**

This paper introduces a DISBench benchmark which evaluates the model's ability on image retrieval with corpus context awareness and a ImageSeeker framework that leverages tools and memories to effectively handle the proposed task. The benchmark is generated through multiple steps of data construction and human verification. In the results, most models present low performance on DISBench, highlighting the benchmark is challenging.

**Compliance With Llm Reviewing Policy:**

Affirmed.

**Final Justification:**

I understand the challenges involved in constructing additional samples for this task, as well as the authors’ explanation regarding agentic retrieval models. While I still encourage the authors to include more samples and incorporate agentic baselines, I have increased my score to weak accept.

**Key Questions For Authors:**

Could authors include more samples to the benchmark?

How other image retireval models perform on DISBench?

**Limitations:**

yes

**Strengths And Weaknesses:**

### Strengths

1. DeepImageSearch introduces a novel task in image retrieval that requires considering corpus-level context. This capability is important and practical for many real-world scenarios.

2. The automatic dataset construction pipeline is well designed and reasonable, helping ensure the quality of the generated queries. In addition, the human verification stage is clearly described in detail.

3. The ImageSeeker framework is evaluated with 10 different backbone models, and the ablation study clearly shows the contribution of each module.

4. The error category distribution analysis highlights potential directions for future research, which is valuable for the research community.

### Weaknesses

1. Limited number of queries in the benchmark. DISBench contains only 122 queries, which is significantly smaller than those in existing image retrieval benchmarks. With such a small number of samples, the variance of the evaluation results may be high, making the results less reliable due to potential fluctuations.

2. Limited evaluation of retrieval models. In the experiments, the base models are evaluated only within the ImageSeeker framework. It would be beneficial to also evaluate other image retrieval methods (including multi-hop or agentic retrieval approaches) to better understand how existing retrieval models perform on this benchmark and task.

3. Limited qualitative examples. Providing more qualitative examples of queries from DISBench would help readers better understand the characteristics and challenges of the benchmark.

---

> ### Author Rebuttal · Authors · 2026-03-31
>
> We sincerely thank the reviewer for recognizing the task novelty, pipeline design, and ablation comprehensiveness. We fully agree that benchmark scale, evaluation breadth, and qualitative illustration are important, and have conducted new experiments during rebuttal to address these concerns.
>
> **Weakness 1 + Question 1: Benchmark Scale**
>
> This is an excellent point and we take it seriously.
>
> **(1) The current scale reflects a deliberate quality–quantity tradeoff.** Each query must contain multiple visually near-identical distractors resolvable only through cross-image contextual reasoning, leading to a 6.1% retention rate. Models solving these hard queries can inherently handle easier variants. The scale is consistent with comparable benchmarks under similar verification (MM-BrowseComp 224, MMSearch 300, MMDR-Bench 140).
>
> **(2) Despite the moderate size, model rankings are statistically stable.** We ran two new analyses. Split-half reliability (10,000 random bisections, ranking all 10 models on each half) yields mean Spearman ρ = **0.911** (5th-percentile 0.806). Bootstrap resampling (10,000 iterations) preserves the top-3 model set in **97.5%** of cases, with clearly separated 95% CIs (Claude-Opus [47.7, 62.2], Gemini-Pro [40.5, 55.2], Claude-Sonnet [36.7, 51.2]). These confirm that the current data reliably distinguishes model capabilities.
>
> **(3) Expansion is actively underway along two complementary directions.** First, we are broadening difficulty coverage: 94 new queries at a moderately relaxed difficulty threshold have already completed Stages 1–3 of the automated pipeline, with only human review remaining. Second, we are onboarding additional users from more diverse demographic and geographic backgrounds to increase corpus variety. Both efforts are ongoing and we expect the benchmark to grow substantially.
>
> ---
>
> **Weakness 2 + Question 2: Evaluation of Retrieval Methods**
>
> We fully agree that evaluating beyond ImageSeeker is important. We now supplement the paper's existing multi-layer evidence with three new experiments.
>
> **(Existing evidence.)** Paradigm level: Table 2 shows SOTA embeddings achieve only 24–30% Recall@10. Model level: Table 1 compares 10 backbones under identical tools. Component level: Table 3 ablations each constitute distinct agent variants.
>
> **(New 1: Unified-metric comparison with embeddings.)** To directly compare under the same EM/F1 metrics, we granted embeddings an **oracle advantage**: the ground-truth target count k per query (agents do not receive this) and computed EM/F1 on top-k results.
>
> | Method | EM | F1 |
> |---|---|---|
> | Qwen3-VL-Emb-2B | 1.6 | 11.3 |
> | Qwen3-VL-Emb-8B | 2.5 | 12.6 |
>
> Even under this unrealistic upper bound, the best F1 is only 12.6—far below the weakest agent (~19). The 2B→8B gain is merely 1.3 F1, confirming a **paradigm-level** bottleneck rather than a capacity bottleneck.
>
> **(New 2: Metadata-augmented embeddings.)** We encoded timestamps and GPS jointly with images. Per-query results: 17% improved, 12% degraded, **71% unchanged**. Enriching single-image representations does not substitute for cross-image reasoning—resolving "beach photos two days after fireworks" requires first locating the fireworks event, which no individual embedding can achieve.
>
> **(New 3: Alternative planning strategy.)** We compared against a **ReAct-style agent** (full tool set, Thought→Action→Observation loops instead of structured decomposition). ReAct is the most widely adopted agentic paradigm and subsumes simpler iterative strategies.
>
> | Strategy | Intra F1 | Inter F1 | Overall F1 |
> |:---|:---:|:---:|:---:|
> | ImageSeeker | 27.1 | 15.5 | 25.4 |
> | ReAct | 21.3 | 10.2 | 19.4 |
>
> With Qwen3-VL-32B, ReAct scores 6.0 F1 lower (24% relative drop), confirming gains from structured decomposition.
>
> ---
>
> **Weakness 3: Qualitative Examples**
>
> We appreciate this suggestion—concrete examples are essential for conveying DISBench's unique challenges. Appendix E provides two full traces (one success, one failure). We supplement with an inter-event case here and will add at least 4 additional annotated examples covering both query types and outcomes in the revision.
>
> > **Query**: *"Find photos containing beer taken during the calendar week when visiting the church whose construction took more than 500 years."*
>
> Reasoning chain: identify the church via world knowledge (e.g., Cologne Cathedral) → locate the user's visit and its date → determine the calendar week → filter for beer photos within that week (11 targets). Claude-Opus found all 11 (F1=1.00); GPT-4o failed to locate the church event (F1=0.00). This illustrates a core challenge: **targets are visually indistinguishable from other dining photos and can only be found through cross-event reasoning**, beyond the reach of embeddings or keyword search.
>
> ---
>
> We sincerely thank the reviewer for the constructive feedback, which has substantially improved the paper. We welcome further discussion.

---

> > ### Author Rebuttal · Reviewer_Ny6X · 2026-04-03
> >
> > I greatly appreciate the authors' effort in the rebuttal. While the rebuttal resovled some of my concerns, I still have some questions and concerns as follows.
> >
> > > **Weakness 1 + Question 1: Benchmark Scale**
> >
> > Thanks for providing additional description and experiment results. However, I still believe that the number of queries (122) is relatively small compared to other benchmarks as the authors mentioned (MM-BrowseComp 224, MMSearch 300, MMDR-Bench 140). Also, while the experimental results indicate that model rankings are relatively stable, this does not fully address my concern regarding the high variance and limited statistical power caused by the small number of queries. In particular, the wide and overlapping confidence intervals suggest that the absolute performance differences between models may not be as clearly distinguishable.
> >
> > As the authors are currently preparing and constructing additional queries, I respectfully recommend that the authors expand the dataset with additional queries and conduct a more comprehensive analysis, then resubmit the work once the evaluation is more robust.
> >
> > > **Weakness 2 + Question 2: Evaluation of Retrieval Methods**
> >
> > Thank you for your explanation. I would like to clarify that my question concerns the evaluation of existing retrieval-based frameworks (including agentic retrieval models) on this benchmark, in order to assess how current state-of-the-art methods (including [1] and [2]) perform.
> > As a benchmark paper, it is important to demonstrate the level of difficulty this task poses for current models, and to verify that the benchmark provides meaningful evaluation that can drive future research on context-aware image retrieval.
> >
> > [1] Zhang, Yifan, et al. "Skywork-R1V4: Toward Agentic Multimodal Intelligence through Interleaved Thinking with Images and DeepResearch." arXiv preprint arXiv:2512.02395 (2025).
> >
> > [2] Zhang, Tao, et al. "MMhops-R1: Multimodal Multi-hop Reasoning." Proceedings of the AAAI Conference on Artificial Intelligence. Vol. 40. No. 33. 2026.

---

> > > ### Author Response · Authors · 2026-04-05
> > >
> > > > **Q1: On Benchmark Scale**
> > >
> > > We thank the reviewer for the continued attention. Due to the word limit of our first-round rebuttal, we could not fully explain these details and clarify below.
> > >
> > > **Benchmark development under this new paradigm proceeds along two directions.** The first is to construct lower-difficulty queries from the existing annotation process, to help the community understand this new task and provide development/debugging data for researchers. The new queries mentioned in our first-round rebuttal belong to this category, and a portion has been completed. The second is to onboard new user albums and discover high-quality queries meeting the test set difficulty standard, a direction worth exploring in the future.
> > >
> > > **What exactly is the difficulty standard of the test set?** The key criterion is whether a query contains sufficient visual distractors that require cross-image reasoning to disambiguate among visually near-identical images. Consider the query "find the Ferris wheel photos from the amusement park trip where we rode the pirate ship." If the user only visited one amusement park, searching "amusement park" directly locates the event with no cross-image reasoning needed. The test set requires scenarios where the user visited four amusement parks with multiple visually similar Ferris wheel photos. The model must first locate pirate ship photos to identify the correct trip, then use temporal constraints to find the Ferris wheel photos from that visit. Mixing in lower-difficulty queries would let frontier models quickly reach near-perfect scores, diluting the benchmark's discriminative power.
> > >
> > > **On the scale of the test set.** Text-based deep search (BrowseComp) and multimodal deep search (MM-BrowseComp 224, MMDR-Bench 140) both remain in the low hundreds despite leveraging the open web. DISBench faces **a dual constraint** beyond these. First, the data source shifts from the open web to closed personal photo albums where models have no prior knowledge of the content. Second, each user's album is limited in size and scenarios meeting the high-difficulty standard are naturally scarce, unlike the open web where exploration is unlimited. **Under these more demanding conditions, DISBench's test set scale is comparable to these peer benchmarks, and this is a necessary stage in the development of such high-difficulty benchmarks.**
> > >
> > > **The core contribution of DISBench lies in defining a new paradigm for image retrieval in the era of agents,** advancing from independent matching to corpus-level contextual reasoning and filling a gap in this direction. This is a nascent research area, and building high-quality benchmarks from closed datasets is itself one of the core challenges. We look forward to the community building on this foundation together.
> > >
> > > ---
> > >
> > > > **Q2: On Evaluation of Existing Agentic Methods**
> > >
> > > We thank the reviewer for the specific suggestions. The Skywork-R1V4 and MMhops-R1 mentioned by the reviewer do possess multimodal agentic capabilities, but **their agentic abilities primarily serve to assist visual reasoning** (e.g., invoking cropping and zooming tools to enhance single-image understanding), and **their retrieval functionality operates in the web search scenario**. DISBench targets a fundamentally different scenario, namely context-aware image retrieval over personal visual histories.
> > >
> > > **Our existing evaluation covers the most frontier models available.** Under the ImageSeeker framework, we evaluated multiple frontier models including Claude-Opus-4.5 and Gemini-3-Pro, representing the current frontier of agentic multimodal reasoning and far stronger than the specialized models mentioned above.
> > >
> > > **To address the reviewer's suggestions, we conducted further investigation and testing during the rebuttal.** Skywork-R1V4's weights are not open-sourced and its API is not accessible. We therefore evaluated its predecessor R1V3 under the same ImageSeeker framework with only the backbone replaced. **The result was EM=0, F1=3.4.** R1V3's 32K context window is insufficient for DISBench, where each retrieval call introduces many images and context compression becomes necessary after one or two rounds, preventing sustained reasoning across steps. MMhops-R1's weights and training code have not been released, making evaluation infeasible.
> > >
> > > **These results further confirm the value of DISBench.** Frontier general-purpose agent models already possess basic context-aware retrieval capabilities in simpler scenarios, while specialized models designed for web search almost entirely fail on this task. Through its abundance of visual distractors and complex cross-image reasoning requirements, the DISBench test set still poses a significant challenge to the strongest available models, demonstrating the research value of this benchmark.
> > >
> > > ---
> > >
> > > We thank the reviewer again for the constructive feedback and hope the above addresses the remaining concerns.

---

### Official Review · Reviewer_LDnd · 2026-03-13

**Soundness:** 2
**Presentation:** 3
**Significance:** 3
**Originality:** 3
**Overall Recommendation:** 4
**Confidence:** 3

**Summary:**

This paper introduces DeepImageSearch, a novel paradigm which summarizes image retrieval as an agentic exploration problem over personal visual histories. The central idea is that many real-world queries cannot be answered by simply treating image-query pairs independently, but instead require reasoning across a sequence of images over temporal, spacial and causal relations. Using a semi-automated pipeline that combines VLM-based context mining and human verification they build DISBench, a benchmark of 122 queries across 57 users (109K photos from YFCC100M). They further implement a baseline agent framework, dubbed ImageSeeker, that integrates retrieval tools with the dual-memory system. Experiments with state-of-the-art multimodal models show that even the best model (Claude-Opus-4. 5) match with ~51% F1, indicating the difficulty of this proposed task.

**Compliance With Llm Reviewing Policy:**

Affirmed.

**Key Questions For Authors:**

1. What is the statistical significance of the performance gaps between models? Given only 122 queries, have you computed confidence intervals or run paired significance tests (e.g., McNemar's test) to verify that the observed differences are statistically meaningful?

2. How would a graph-based approach that directly operates on your memory graph (e.g., using graph traversal or GNN-based reasoning) compare with the tool-use agent paradigm? Your benchmark construction builds explicit memory graphs, yet the evaluation completely ignores graph-based methods.

3. Since both the query synthesis (Gemini-3-Pro) and the evaluation (Gemini-3-Pro-Preview) involve the same model family, how do you control for potential data leakage or distributional bias? Have you verified that the synthesized queries do not favor specific model architectures?

4. How sensitive are the results to the choice of embedding model for ImageSearch? Table 1 shows results for Qwen3-VL-Embedding 2B and 8B, but the performance differences are inconsistent across backbone models. A more systematic study of the retrieval-reasoning interaction would strengthen the paper.

5. Could you provide a more detailed breakdown of performance by query complexity (e.g., number of reasoning hops, collection size, temporal span), rather than just the coarse intra/inter-event split?

**Strengths And Weaknesses:**

### Strengths

1. Well-motivated problem formulation. The paper well illustrates an important gap between independent semantic matching and the context-dependent reasoning needed in real-world photo search. Readers should be able to intuitively understand the intra-event vs inter-event query difference and should recognize it as a true retrieval challenge.

2. Careful benchmark construction pipeline. This semi-automated pipeline (VLM-driven association mining + human verification at inter-annotator IoU 0.91) is a pragmatic solution to generating reasoning-heavy queries at affordable price. The moat of the quality control criteria (visual ambiguity, contextual identifiability, strong-to-weak reasoning flow) is conceptually sound.

3. Comprehensive evaluation across diverse models. The paper provides informative comparisons, benchmarking many proprietary and open-source models in a protected environment with the same tool interfaces. The error analysis and test-time scaling experiments are useful diagnostic insights.


### Weaknesses

1. Small benchmark scale raises concerns of statistical reliability. The benchmark has just 122 queries within 57 users, a 6.1% retention rate from 2,000 candidates. Although the authors note this as a limitation, it does raise me concern about (a) the statistical power of any differences reported between models, particularly in cases where those differences are small, and (b) potential bias in query distribution retentive. No significance analysis (e.g., CI or bootstrap tests) is performed, which the authors miss. Concurrent with its introduction, MR^2-Bench [1] is a highly reasoning-intensive multimodal retrieval benchmark engineered from the ground-up to be cost-effective to crowdsourced over 1000 queries over hundreds of diverse reasoning types. Though still an order of magnitude more complex than existing pipelines, this definitively signals that there are at least more scalable ways to construct such challenges upfront while maintaining query quality.
2. Not a technology-intensive contribution of the agent framework. Seeker is basically a standard LLM tool-use agent with domain specific tools and a separation of session / working memory. The dual-memory mechanism (where history is compressed to session memory and used as working memory) closely follows prior context-management strategies for long-horizon agents [2]. The paper emphasizes the framework serves as a “baseline” but does not address whether its design choices (i.e., the exact tool set, subset-backed state management) are ideal or just pragmatic. It compares with no alternative agent architectures (e.g., ReAct, tree-of-thought planning or reflection-based agents). The authors should compare with strands of recent work in agentic retrieval frameworks such as MC-Search [3], which also assesses multimodal agentic search but goes further to offer step-wise process annotations and process-level metrics.
3. Missing important concurrent baselines. A number of concurrent works tackle closely related problems, which are neglected by authors: (a) PhotoBench [4] also creates a personal album retrieval benchmark from real-life metadata rich collections; (b) MC-Search [3] evaluates multimodal agentic RAG with structured reasoning chains and step-wise evaluation; (c) MR^2-Bench [1] benchmarks reasoning intensive multimodal retrieval. These are not mentioned or compared with in the paper. Additionally, no graph-based retrieval methods are established as baselines even though the paper's own memory graph representation inherently lends itself to graph traversal or graph neural network based approaches.
4. Evaluation lacks process-level assessment. Because the task is agentic and multi-step, simply assessing the end output (EM and F1) is not enough. It is still open question whether—when models achieve similar F1 score, they come to their conclusions by similar reasoning paths or genuinely different strategies. The manual error analysis (Section 3.9) is performed on "sampled failure cases" without any mention of sample size or entropy, nor their selection process, making it hard to reproduce these results. The authors may want to employ process-level metrics (like intermediate precision at every reasoning hop) like MC-Search [3].
5. Circular evaluation concern. The benchmark construction pipeline is based on VLMs (Qwen3-VL-235B for parsing and Gemini3-Pro for query synthesis), and then evaluate models in the same families via the constructed benchmark. This now has me questioning if there is an implicit bias in the query distribution towards reasoning styles that these families of VLM handle in well (or poorly) and whether a benchmark really captures actual retrieval needs from users or patterns synthesized by VLMs reasoning. The paper has no discussion of this potential confound.

## References

[1]Zhou J, Liu Z, Xiong L, et al. MR $^ 2$-Bench: Going Beyond Matching to Reasoning in Multimodal Retrieval[J]. arXiv preprint arXiv:2509.26378, 2025.

[2] Yao H, Zhang R, Huang J, et al. A survey on agentic multimodal large language models[J]. arXiv preprint arXiv:2510.10991, 2025.

[3] Ning X, Fu D, Wei T, et al. MC-Search: Evaluating and Enhancing Multimodal Agentic Search with Structured Long Reasoning Chains[J]. arXiv preprint arXiv:2603.00873, 2026.

[4] Xu T, Shan R, Wu J, et al. PhotoBench: Beyond Visual Matching Towards Personalized Intent-Driven Photo Retrieval[J]. arXiv preprint arXiv:2603.01493, 2026.

---

> ### Author Rebuttal · Authors · 2026-03-31
>
> We sincerely thank the reviewer for recognizing the problem formulation, pipeline design, and evaluation comprehensiveness. We conducted new experiments during rebuttal to address the concerns.
>
> **W1+Q1+Q5: Benchmark Scale and Statistical Reliability**
>
> **(1)** To ensure DISBench can guide model development over an extended period, we set a high quality bar where each query requires cross-image contextual reasoning that cannot be bypassed by appearance matching. Under these criteria we produced 122 high-quality queries and will continue expanding. This scale is consistent with comparable multimodal benchmarks (MM-BrowseComp 224, MMSearch 300, MMDR-Bench 140).
>
> **(2) Model rankings are statistically stable.** Split-half reliability (10,000 bisections) yields mean Spearman ρ = **0.911** (ρ ≥ 0.806 in 95% of splits). Bootstrap (10,000 resamples) keeps the top-3 set unchanged in **97.5%** of cases, with separated 95% CIs: Claude-Opus [47.7, 62.2], Gemini-Pro [40.5, 55.2], Claude-Sonnet [36.7, 51.2].
>
> **(3) Q5.** Beyond the coarse intra/inter split, we added a breakdown in the Appendix stratifying performance by target count and reasoning pattern. Key findings: (a) all strong models show a consistent 9–14 F1 gap between intra- and inter-event queries, confirming cross-event reasoning as the shared bottleneck; (b) on single-target queries, all models except Claude-Opus plateau around F1 30, showing even the simplest subset remains challenging.
>
> **(4) Expansion is underway along two directions.** First, broadening difficulty: **94 new queries** at a relaxed threshold have completed the automated pipeline (Stages 1–3), with only human review remaining. Second, onboarding users from more diverse demographic and geographic backgrounds.
>
> ---
>
> **W2+Q4: Agent Framework**
>
> **(1) ImageSeeker is a deliberate baseline, not a methodological contribution.** DeepImageSearch defines a new task where no existing framework applies directly—the required tools (subset-backed retrieval, metadata filtering, visual verification over photo histories) differ fundamentally from web or GUI agents. We kept ImageSeeker simple and modular so the benchmark serves as the lasting contribution.
>
> **(2)** Beyond this baseline, the paper offers multi-layer evidence: Table 2 shows the embedding ceiling (Recall@10 only 24–30%), Table 1 compares 10 backbones under identical tools to isolate reasoning, and Table 3's ablations constitute distinct agent variants.
>
> **(3)** To address the lack of alternative planning strategies, we added a **ReAct baseline** (full tool set, Thought→Action→Observation loops):
>
> | Method | Backbone | EM | F1 |
> |--------|----------|-----|------|
> | ReAct | Qwen3-VL-32B | 5.7 | 19.4 |
> | ImageSeeker | Qwen3-VL-32B | 10.7 | 25.4 |
>
> A 24% relative F1 drop confirms gains from structured decomposition.
>
> **Q4.** Claude-Opus achieves nearly identical F1 under 2B and 8B embeddings (gap only 0.5), while backbone gaps under the same embedding exceed 33 F1 points. The bottleneck lies in reasoning, not retrieval quality.
>
> ---
>
> **W3+Q2: Concurrent Works and Graph-based Methods**
>
> We appreciate the pointers. These works appeared around the same time as our submission and we will add a thorough discussion. PhotoBench focuses on single-image personalized intent without corpus-level reasoning; MR²-Bench scores each image independently; MC-Search assumes a unique fixed reasoning chain, whereas DISBench queries admit multiple valid paths.
>
> **Q2.** The memory graph serves as construction-stage infrastructure and is not exposed at evaluation. Exploring graph-based reasoning at inference is a promising direction we plan to investigate.
>
> ---
>
> **W4: Process-level Evaluation**
>
> We apologize for misleading wording in Section 5.6. The error analysis was conducted **exhaustively over all failed traces** for Claude-Opus (51 traces), not on a sample. We will correct this.
>
> We agree that process-level evaluation is valuable. A core challenge is that each query admits multiple valid paths, making a single ground-truth trajectory hard to define. We instead analyzed strategies directly: top-3 models use 2.3× turns and 3.2× tool calls, favoring AlbumSearch+ViewPhotos (each 9pp higher), while weaker models over-rely on FilterMetadata (12pp higher). We will explore systematic process metrics in future work.
>
> ---
>
> **W5+Q3: Circular Evaluation**
>
> **Our strongest evidence: Claude-Opus-4.5, entirely uninvolved in any construction stage, achieves the best F1.** Qwen3-VL-235B and Gemini-3-Pro (used in construction) score lower. If systematic bias favored construction models, we would expect the opposite.
>
> Construction and evaluation test different skills: construction VLMs perform visual parsing and query generation, while evaluation requires multi-step planning and long-horizon tool coordination. VLM outputs are only the starting point, with 93.9% filtered through four human stages.
>
> ---
>
> We thank the reviewer again and welcome further discussion.

---

> > ### Author Rebuttal · Reviewer_LDnd · 2026-04-03
> >
> > I thank the authors for the detailed rebuttal and new experiments. The statistical reliability analysis (split-half ρ = 0.911), ReAct baseline comparison, and circular evaluation clarification have addressed my main concerns.
> >
> >  Remaining minor concerns:
> >  - I still encourage the authors to include a discussion of PhotoBench, MR²-Bench, and MC-Search in the related work, as promised.
> >  - The expansion plan (94 new queries) is encouraging; I hope the camera-ready can report updated scale.
> >
> > I encourage the authors to include the discussed concurrent works and updated benchmark scale in the camera-ready. I maintain my score.

---

> > > ### Author Response · Authors · 2026-04-05
> > >
> > > We thank the reviewer for the continued feedback and support. We address the two remaining questions below.
> > >
> > > > **Q1: On the discussion of concurrent work**
> > >
> > > In our first-round rebuttal we briefly described the key task-level differences between PhotoBench, MR²-Bench, MC-Search and DISBench. We elaborate here. PhotoBench performs multi-source constraint matching over authentic personal albums, but focuses on personalized intent satisfaction for individual images without corpus-level cross-image reasoning. MR²-Bench evaluates reasoning-intensive retrieval, but scores each document independently without requiring understanding of inter-document relationships. MC-Search prescribes a single fixed reasoning chain per query, whereas DISBench queries admit multiple valid reasoning paths. After detailed analysis, we confirm that these works and DISBench address distinct research problems with clearly different task definitions. **We will provide a more thorough analysis of these works in the camera-ready, clarifying their respective task scopes and complementary relationships.**
> > >
> > > > **Q2: On benchmark scale**
> > >
> > > Benchmark development under this new paradigm proceeds along two directions. The first is to construct lower-difficulty queries from the existing annotation process, to help the community understand this new task and provide development/debugging data for researchers. The new queries mentioned in our first-round rebuttal belong to this category, and a portion has been completed. The second is to onboard new user albums and discover high-quality queries meeting the test set difficulty standard, a direction worth exploring in the future. The test set itself maintains a strict difficulty standard, ensuring that every query contains sufficient visual distractors and **must be resolved through cross-image reasoning to disambiguate**, preventing frontier models from quickly closing the benchmark.
> > >
> > > ---
> > >
> > > We thank the reviewer again for the constructive feedback and will incorporate the above improvements in the camera-ready. We hope this response fully addresses the remaining concerns.

---

### Decision · Program_Chairs · 2026-04-30

**Decision:**

Accept (regular)

**Comment:**

The article received unanimous positive recommendations, with the reviewers appreciating the problem formulation (LDnd, Ny6X, sLBf), the proposed benchmark (LDnd, Ny6X, sLBf), as well as the comprehensive evaluation (LDnd, sLBf) and analyses (Ny6X). Initial concerns, e.g., benchmark scale (LDnd, Ny6X), baselines (LDnd, Ny6X), qualitatives (LDnd, Ny6X, b2G4), and metrics (sLBf) were mostly addressed by the rebuttal.

The AC went through the manuscript, reviews, and rebuttal, finding no reasons to overturn the consensus. The promised clarifications and additional analyses should be included in the final version of the manuscript.